# GAS41 modulates ferroptosis by anchoring NRF2 on chromatin

Zhe Wang[1,9], Xin Yang[1,9], Delin Chen[1], Yanqing Liu[1], Zhiming Li [1], Shoufu Duan[1], Zhiguo Zhang [1,2,3,4], Xuejun Jiang [5], Brent R. Stockwell [6,7] & Wei Gu [1,2,8] ✉

YEATS domain-containing protein GAS41 is a histone reader and oncogene. Here, through genome-wide CRISPR-Cas9 screenings, we identify GAS41 as a repressor of ferroptosis. GAS41 interacts with NRF2 and is critical for NRF2 to activate its targets such as SLC7A11 for modulating ferroptosis. By recognizing the H3K27-acetylation (H3K27-ac) marker, GAS41 is recruited to the *SLC7A11* promoter, independent of NRF2 binding. By bridging the interaction between NRF2 and the H3K27-ac marker, GAS41 acts as an anchor for NRF2 on chromatin in a promoter-specific manner for transcriptional activation. Moreover, the GAS41-mediated effect on ferroptosis contributes to its oncogenic role in vivo. These data demonstrate that GAS41 is a target for modulating tumor growth through ferroptosis. Our study reveals a mechanism for GAS41-mediated regulation in transcription by anchoring NRF2 on chromatin, and provides a model in which the DNA binding activity on chromatin by transcriptional factors (NRF2) can be directly regulated by histone markers (H3K27-ac).

Proteins participating in post-translational histone modifications are critical transcriptional machinery components, which have been categorized as readers, movers, writers, or erasers. YEATS domain-containing proteins are identified as epigenetic readers, including four evolutionary conserved proteins in humans: Eleven-Nineteen Leukemia (ENL, encoded by *YEATS1*), YEATS2, ALL1-Fused gene from chromosome 9 protein (AF9, encoded by *YEATS3*), and Glioma Amplified Sequence 41 (GAS41, encoded by *YEATS4*). Beyond the initial identification of GAS41 as a frequently amplified gene in glioblastoma, amplification of GAS41 and abnormal upregulated expression levels of GAS41 are also found in a variety of human cancers, including sarcoma, lung, bladder, and uterine cancers[1–4]. Specifically, abrogation of GAS41 expression has been reported to suppress non-small cell lung cancer (NSCLC) cell growth and survival through regulating DNA replication

and cell cycle[4]. Dysregulation of GAS41 is also associated with gastric carcinoma, hepatic carcinoma, breast cancer, colorectal cancer, and pancreatic cancer[5–9]. In addition, GAS41 plays a critical role in innate lymphoid cell lineage commitment, which defends infections and maintains mucosal homeostasis[10]. GAS41 is also required for the maintenance of embryonic stem cell identity[11]. Thus, as an oncogene, targeting GAS41 is a promising therapeutic method for various diseases. However, the functions of GAS41 in tumorigenesis processes have not yet been fully revealed.

YEATS domain-containing proteins recognize diverse lysine modification[12], including acetylation, butyrylation, crotonylation, propionylation, and succinylation, which are highly enriched in transcription-modulating chromatin[4,13–18]. Each of them is involved in the fundamental process of transcriptional regulation, such as

[1]Institute for Cancer Genetics, Vagelos College of Physicians & Surgeons, Columbia University, New York, NY, USA. [2]Herbert Irving Comprehensive Cancer Center, Vagelos College of Physicians & Surgeons, Columbia University, New York, NY, USA. [3]Department of Pediatrics, Vagelos College of Physicians & Surgeons, Columbia University, New York, NY, USA. [4]Department of Genetics and Development, Vagelos College of Physicians & Surgeons, Columbia University, New York, NY, USA. [5]Cell Biology Program, Memorial Sloan-Kettering Cancer Center, New York, NY, USA. [6]Department of Chemistry, Columbia University, New York, NY, USA. [7]Department of Biological Sciences, Columbia University, New York, NY, USA. [8]Department of Pathology and Cell Biology, Vagelos College of Physicians & Surgeons, Columbia University, New York, NY, USA. [9]These authors contributed equally: Zhe Wang, Xin Yang. ✉e-mail: wg8@cumc.columbia.edu

chromatin structure and gene transcription[4,13,15–17]. GAS41 plays a role as the component of SNF2-related CREBBP activator protein (SRCAP) and TIP60/p400 chromatin remodeling complexes[19,20]. Recognition of histone acetylation by GAS41 promotes the exchange of canonical histone H2A for the H2A.Z variant catalyzed by TIP60/p400 and SRCAP complexes in specific chromatin[4]. GAS41 activates Lmo4 transcription through H3K27 acetylation by binding to Dot1l-RNA Pol II complex[10]. MYC recruits GAS41/SIN3A-HDAC1 complex to repress gene expression in chromatin-mediated by H3K27 crotonylation[21]. On the other hand, GAS41 also directly interacts with transcriptional factor AP-2β, the mixed lineage leukemia (MLL) fusion partner AF10, and a subunit of TFIIF, the RAP30[22–24].

Here, from our genome-wide CRISPR-Cas9 screen, we identified *YEATS4* as a regulator for ferroptosis defense. Depletion of GAS41 sensitized NSCLC cells to ferroptotic cell death. Mechanistically, GAS41 interacted with NRF2 and promoted NRF2 transcriptional ability, specifically on glutathione (GSH) metabolism genes, *SLC7A11* and *GCLC*. Our study underlines a mechanism for NRF2 transcriptional regulation and provides a potential method for NSCLC therapy through targeting GAS41-mediated ferroptosis defense.

## Results

### Identification of GAS41 as a ferroptosis repressor upon ROS stress
Iron-dependent ROS accretion initiates ferroptosis, which is a form of programmed cell death triggered by an overload of lipid peroxidation on the cellular membrane. We have established that high levels of ROS generated by tert-butyl hydroperoxide (TBH), an organic peroxide ROS generator, can induce p53-dependent ferroptosis, regardless of ACSL4 status[25–28]. Thus, to uncover the potential regulators that modulate cell sensitivity to ferroptosis induced by ROS accumulation, we conducted a genomic-wide CRISPR-Cas9 screen in human melanoma A375 cells treated with TBH. sgRNA abundance sequencing was analyzed by the Model-based Analysis of Genome-wide CRISPR-Cas9 Knockout (MAGeCK) algorithm[29]. As shown in Fig. 1a, known regulators that promote ferroptosis appear in top positive selection targets, such as *KEAP1*, *POR*, and *RETSAT*[30–33]. Among the genes that being knocked out induced ferroptosis, the top hits included several well-known ferroptosis suppressors, such as *AIFM2* (encodes FSP1), *GSS*, *SLC7A11*, and *GCLM*, which proves the robustness of the screen[27,34,35] (Fig. 1a and Supplementary Fig. 1a). *YEATS4*, with the similar β score as *SLC7A11* (*YEATS4*, β score −0.58, *p* value 0.00068 vs *SLC7A11*, β score −0.47, *p* value 0.00318), was the top hit which is required for cell survival upon TBH treatment.

To validate our result of the CRISPR-Cas9 screen, we first transfected A375 cells with non-targeting negative control sgRNA (sgNC) or two individuals sgRNAs targeting GAS41 (sgGAS41#1 and sgGAS41#2). The expression of SLC7A11, a critical p53-mediated transcriptional repression metabolic target, was downregulated upon GAS41 knockdown[27] (Fig. 1b). Moreover, we examined the possibility of GAS41 in regulating several well-known factors in ferroptosis defense, including GPX4, FSP1, and DHODH. Unlike SLC7A11, loss of GAS41 had no obvious effect on the expression of GPX4, FSP1, and DHODH (Supplementary Fig. 1b), indicating that SLC7A11 could be the major regulated target upon loss of GAS41 in ferroptosis defense.

Then, we examined the cell viability upon common ferroptosis inducers. Indeed, GAS41 knockdown increased the cellular vulnerability to ferroptosis induced by TBH or imidazole ketone erastin (IKE), a potent inhibitor of SLC7A11 to inhibit cystine import and GSH synthesis (Fig. 1c, d). Although GAS41 was initially identified as an amplified gene in glioma, recent studies found that GAS41 amplification accompanies the upregulation of GAS41 expression in various cancer types, especially in NSCLC[1,4,36]. Thus, to further explore the potential role of GAS41 in ferroptotic responses, we knocked out GAS41 in two additional NSCLC cell lines, A549 and H460.

Consistently, GAS41 deficiency sensitized A549 and H460 cells to TBH-induced ferroptotic cell death with increased levels of lipid peroxidation (Fig. 1e–j). Moreover, loss of GAS41 also sensitized cells to ferroptosis induced by RSL-3 treatment, IKE treatment, and cystine starvation (Fig. 1k and Supplementary Fig. 1c–g), suggesting that GAS41 is a universal ferroptosis suppressor. Collectively, our data indicate that GAS41, an epigenetic regulator, acts as a ferroptosis suppressor.

### GAS41 suppresses ferroptosis in a p53-independent manner
Given that p53 activation is critical for ferroptosis upon ROS-induced oxidative stress and GAS41 has been reported to be involved in p53 transcriptional regulation[36,37], we then evaluated whether GAS41 is involved in p53-mediated ferroptosis. To this end, A549 sgNC and sgGAS41 cells were treated with or without Nutlin-3. As shown in Supplementary Fig. 2a, p53 activation induced by Nutlin-3 treatment resulted in the decrease of SLC7A11 expression; loss of GAS41 reduced SLC7A11 expression independent of p53 activity (Supplementary Fig. 2a). We observed that p53 activation sensitized cells to ferroptosis (Supplementary Fig. 2b, blue line vs red line), however, the levels of ferroptotic cell death were again upregulated by GAS41 knockout upon Nutlin-3 treatment (Supplementary Fig. 2b, orange line and purple line vs blue line). In parallel, lipid peroxidation induced by GAS41 knockout was again increased in the presence of Nutlin-3 (Supplementary Fig. 2c). To further evaluate the role of GAS41 in p53-mediated ferroptosis, we knocked down GAS41 in both H1299 cells (p53-null NSCLC cell line) and A375 p53-null cells. GAS41 knockdown decreased the cell viability of H1299 cells upon TBH or IKE treatment (Supplementary Fig. 2d–f). Similar results were obtained in A375 p53 knockout cells (Supplementary Fig. 2g–i). Altogether, loss of GAS41 promotes ferroptosis vulnerabilities upon ROS accumulation in a p53-independent manner.

### GAS41 interacts NRF2 in vitro and in vivo
As an epigenetic regulator, GAS41 is a subunit for both SRCAP and TIP60/p400 chromatin remodeling complexes, which can catalyze the deposition of histone variant H2A.Z into chromatin[4,11]. GAS41 is also recruited by some transcription factors to act as a co-factor, for example, TFIIF subunits RAP30 and AP-2β[22,23]. However, GAS41 has no DNA binding affinity and preferentially acts as a histone acetylation reader[4]. Since GAS41 knockdown appreciably affects SLC7A11 expression in a p53-independent manner, it is conceivable that GAS41-mediated ferroptosis resistance by regulating other critical transcription factors involving antioxidant modulation. Of note, we identified 12 unique peptides of GAS41 from the NRF2-associated protein complex[38] (Fig. 2a, b and Supplementary Fig. 3a), suggesting GAS41 as a potential NRF2-interacting protein. To verify the interaction between NRF2 and GAS41, we first established SFB-GAS41 stable expressing HEK293T cell line and purified GAS41 interacting proteins through a two-step affinity purification protocol (first with streptavidin agarose beads and then with S-protein agarose beads). As shown in Fig. 2c, GAS41 interacted with the components of the TIP60/TRRAP complex, which proves the reliability of our SFB-GAS41 stable cell line. As expected, NRF2 was readily detected in the immunoprecipitated complexes of GAS41 under the same condition (Fig. 2c). Next, we transfected HEK293T cells with NRF2 or GAS41 expressing plasmids alone or co-transfected HEK293T cells with NRF2 and GAS41 expressing plasmids. NRF2 was detected in the immunoprecipitated complexes of GAS41 (Fig. 2d), and vice versa (Supplementary Fig. 3b). To further prove this point, we examined the interaction between NRF2 and GAS41 under physiological conditions. In A549 cells, endogenous NRF2 was detected from GAS41 immunoprecipitated complexes but not IgG control (Fig. 2e). Consistently, GAS41 was also detected from endogenous NRF2 interacting proteome (Fig. 2f).

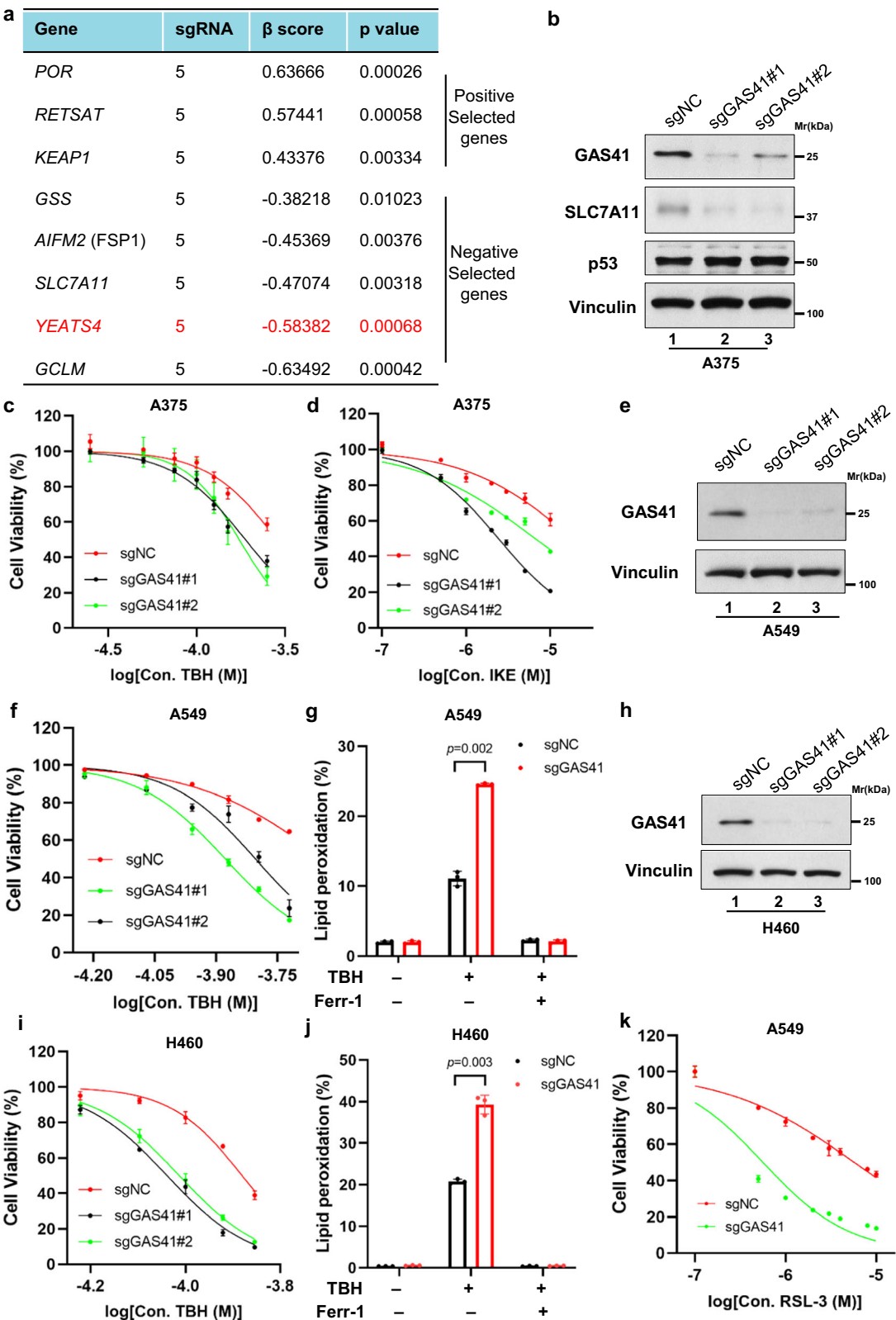

To elucidate the direct interaction of NRF2 and GAS41, we mapped the binding of GAS41 to GST-fused NRF2 isoforms in vitro (Supplementary Fig. 3c). As shown in Supplementary Fig. 3d, GAS41 directly bound with Neh1 and Neh3 domains of NRF2 (amino acids, AA. 434–605), but not GST alone, suggesting the domains responsible for Antioxidant Response Element (ARE) recognition and transactivation of NRF2 is also critical for NRF2 binding to GAS41[39,40]. We then divided GAS41 into two parts: N-terminal (AA. 1–160), which contains YEATS domain for reading histone acetylation, and C-terminal (AA. 161–227), which contains coiled-coil regions (Fig. 2g). NRF2 interacted with GAS41 C-terminus, but not GAS41 N-terminus, potentizing the scaffolding role of GAS41 in bridging NRF2 to histone modification markers (Fig. 2h). Moreover, loss of AA. 184–227 or 207–227 within the C-terminal of GAS41 both abrogated the binding between NRF2 to

**Fig. 1 | Identification of GAS41 as a ferroptosis repressor upon ROS stress. a** Top hit of genomic-wide CRISPR-Cas9 screen in human melanoma A375 cells treated with TBH and Nutlin-3 were shown as a table. *YEATS4* was highlighted in red. **b** Western blot of GAS41 and SLC7A11 protein levels in sgNC and sgGAS41 A375 cells generated using control sgRNA and two individuals targeting GAS41 sgRNA, respectively. **c** Cell viability of sgNC and sgGAS41 A375 cells treated with TBH for 6 h. **d** Cell viability of sgNC and sgGAS41 A375 cells treated with IKE for 24 h. **e** Western blot of GAS41 protein levels of sgNC and sgGAS41 A549 cells generated using control sgRNA and two individuals targeting GAS41 sgRNAs, respectively. **f** Cell viability of sgNC and sgGAS41 A549 cells treated with TBH for 6 h. **g** Assessment of lipid peroxidation by flow cytometry after C11-BODIPY staining of

sgNC and sgGAS41 A549 cells treated with TBH (120 μM) and Ferr-1 (2 μM) for 4 h as indicated. **h** Western blot of GAS41 protein levels in sgNC and sgGAS41 H460 cells generated using control sgRNA and two individuals targeting GAS41 sgRNA, respectively. **i** Cell viability of sgNC and sgGAS41 H460 cells treated with TBH for 6 h. **j** Assessment of lipid peroxidation by flow cytometry after C11-BODIPY staining of sgNC and sgGAS41 H460 cells treated with (120 μM) and Ferr-1 (2 μM) for 4 h as indicated. **k** Cell viability of sgNC and sgGAS41 A549 cells treated with RSL-3 for 24 h. Data are mean ± SD of $n = 3$ independent biological repeats. *p* values were calculated using unpaired, two-tailed Student's *t* test. Western blot experiments above (**b**, **e**, and **h**) were repeated three times with similar results and representative images are shown. Source data are provided as a Source Data file.

GAS41, indicating the integrity of the GAS41 coiled-coil region is required for the interaction (Fig. 2i). Taken together, our results show that GAS41 is a bona fide binding partner of NRF2 both in vivo and in vitro.

## GAS41 promotes NRF2 transcriptional ability on antioxidant genes

Given the interaction between GAS41 and NRF2, we hypothesized that GAS41-mediated ferroptosis resistance is through modulating NRF2-dependent antioxidant function. To this end, we first examined the effects of GAS41 on NRF2 protein levels. As shown in Fig. 3a, the loss of GAS41 had no effects on NRF2 protein expression levels. We next investigated whether GAS41 modulates NRF2 transcriptional ability. To this end, we examined the expression levels of NQO1, SLC7A11, and GCLC in both KEAP1 mutated (A549 and H460) or wild-type (H1299) NSCLC cells. Significantly downregulated NQO1, SLC7A11, and GCLC mRNA and protein levels were observed after loss of GAS41 regardless of *KEAP1* status, which is in accordance with our hypothesis (Fig. 3a–d). Consistent with the above results, Tet-On-induced GAS41 knockdown in A549 cells significantly suppressed the mRNA level and protein levels of NQO1, SLC7A11, and GCLC (Fig. 3e and Supplementary Fig. 4a). Of note, GAS41 knockdown resulted in a decrease in SLC7A11 and GCLC expression level was both found with or without tert-butylhydroquinone (tBHQ) treatment, which is a potent NRF2 activator through abrogating KEAP1/NRF2 interaction and KEAP1-mediated NRF2 ubiquitination (Fig. 3f, g), indicating that GAS41 is required for NRF2 transcriptional activation. To further prove this point, luciferase reporter containing the promoter sequences of *SLC7A11* alone or along with dose-dependent overexpression of NRF2 expressing vector was transfected into A549 sgNC and sgGAS41 cells. As shown in Fig. 3h, compared with A549 sgNC cells, GAS41 knockout significantly downregulated the luciferase activity of a *SLC7A11* reporter in the presence of equal expression levels of NRF2. Taken together, our data demonstrate that GAS41-mediated ferroptosis defense is through modulating NRF2 transcriptional abilities.

Next, to elucidate the role of these NRF2-target genes regulated by GAS41 in ferroptosis defense, we restored SLC7A11, GCLC, and NQO1 expression in A549 sgGAS41 cells, respectively, and tested these cells sensitivity to ferroptosis. As shown in Supplementary Fig. 4b–g, individually overexpressed SLC7A11, GCLC, and NQO1 can partially rescue ferroptosis resistance in GAS41 knockout cells. These results not only proved the effects of SLC7A11, GCLC, and NQO1 against oxidative stresses and ferroptosis but also emphasized the potential function of GAS41 in ferroptosis by modulating these three antioxidative genes.

## The interaction between GAS41 and NRF2 is required for NRF2 transcriptional ability

Given that GAS41 and NRF2 interact with each other, it is convincible that this interaction could play critical roles in GAS41-mediated NRF2 transcription. To identify the key amino acid(s) responsible for GAS41 binding with NRF2, we screened several mutants located in AA. 1–160 or AA. 207–227 which was necessary for histone acetylation-GAS41

binding or NRF2-GAS41 interaction, respectively (Fig. 2h, i)[4]. Thus, we examined the interaction between NRF2 and GAS41 point mutants. Indeed, one point mutant of GAS41 (L211A) almost abolished the binding with NRF2, compared with GAS41 wildtype (WT) (Fig. 4a). In addition, GAS41 W93A mutant has been reported that lost the ability to bind with histone acetylation[4]. We further found that GAS41 W93A retained its ability to interact with NRF2 (Fig. 4a). Therefore, we selected GAS41 W93A and L211A mutants that specifically abrogated the binding with histone acetylation or NRF2, respectively, to elucidate the effect of NRF2-GAS41 interaction on NRF2 transcriptional ability.

Next, GAS41 WT, GAS41-W93A, and GAS41-L211A were expressed in GAS41-null A549 cells to dissect the effect of GAS41 on modulating NRF2 transcriptional functions. As shown in Fig. 4b, western blot analysis revealed that GAS41 WT, but not either histone acetylation binding-deficient mutant (W93A) or NRF2 binding-deficient mutant (L211A), is able to restore the ability of NRF2-mediated transcriptional activation of NQO1, SLC7A11, and GCLC in GAS41-null cells. Similar results were also obtained for the mRNA levels of those NRF2-targets by qPCR-analysis (Fig. 4c and Supplementary Fig. 5a). Moreover, GAS41 WT but not GAS41-W93A, or GAS41-L211A, was able to increase the GSH levels in GAS41-null cells (Fig. 4d), indicating that both the GAS41-NRF2 binding, and the GAS41-histone acetylation interaction are required for NRF2 transcriptional activation mediated by GAS41.

Finally, we performed ferroptosis assays with those point mutants. As expected, GAS41 WT but not GAS41-W93A or GAS41-L211A, was able to suppress ferroptosis in GAS41-null cells (Fig. 4e). In parallel, lipid peroxidation induced by GAS41 deficiency was downregulated by overexpressed GAS41 WT, but not GAS41 W93A or L211A mutants (Fig. 4f, g). Taken together these data validate that both the GAS41-NRF2 interaction and the GAS41-histone acetylation binding are critical for GAS41-mediated effects on NRF2 transactivation and ferroptosis defense.

## The regulatory interplay of NRF2 and GAS41 in transcriptional regulation

Our aforementioned results suggest that GAS41 interacts with NRF2 to transcriptionally regulate NRF2-dependent antioxidant gene expression; however, as an epigenetic reader, how GAS41 implements this regulation requires further investigation. Previous studies showed that the recognition of GAS41 to acetylated histones helps the association of H2A.Z deposited by SRCAP and TIP60/p400 chromatin remodeling complexes with chromatin[4,11], it is possible that the transcriptional regulation of GAS41 on NRF2-target genes is also dependent on chromatin remodeling complexes. To the end, we knocked down the critical subunits, TIP60 and TRRAP, among the TIP60/TRRAP complex by small interfering RNA (siRNA) technique and tested the changes of antioxidant genes. The mRNA levels of *NQO1*, *SLC7A11*, and *GCLC* showed no obvious change by knockdown of either TIP60 or TRRAP (Fig. 5a), suggesting that neither TIP60 nor TRRAP is involved in GAS41-NRF2 dependent transcriptional regulation. Interestingly, as shown in Supplementary Fig. 6a, b, we found that the GAS41 YEATS domain (AA. 1–160) was the critical region for the interaction between

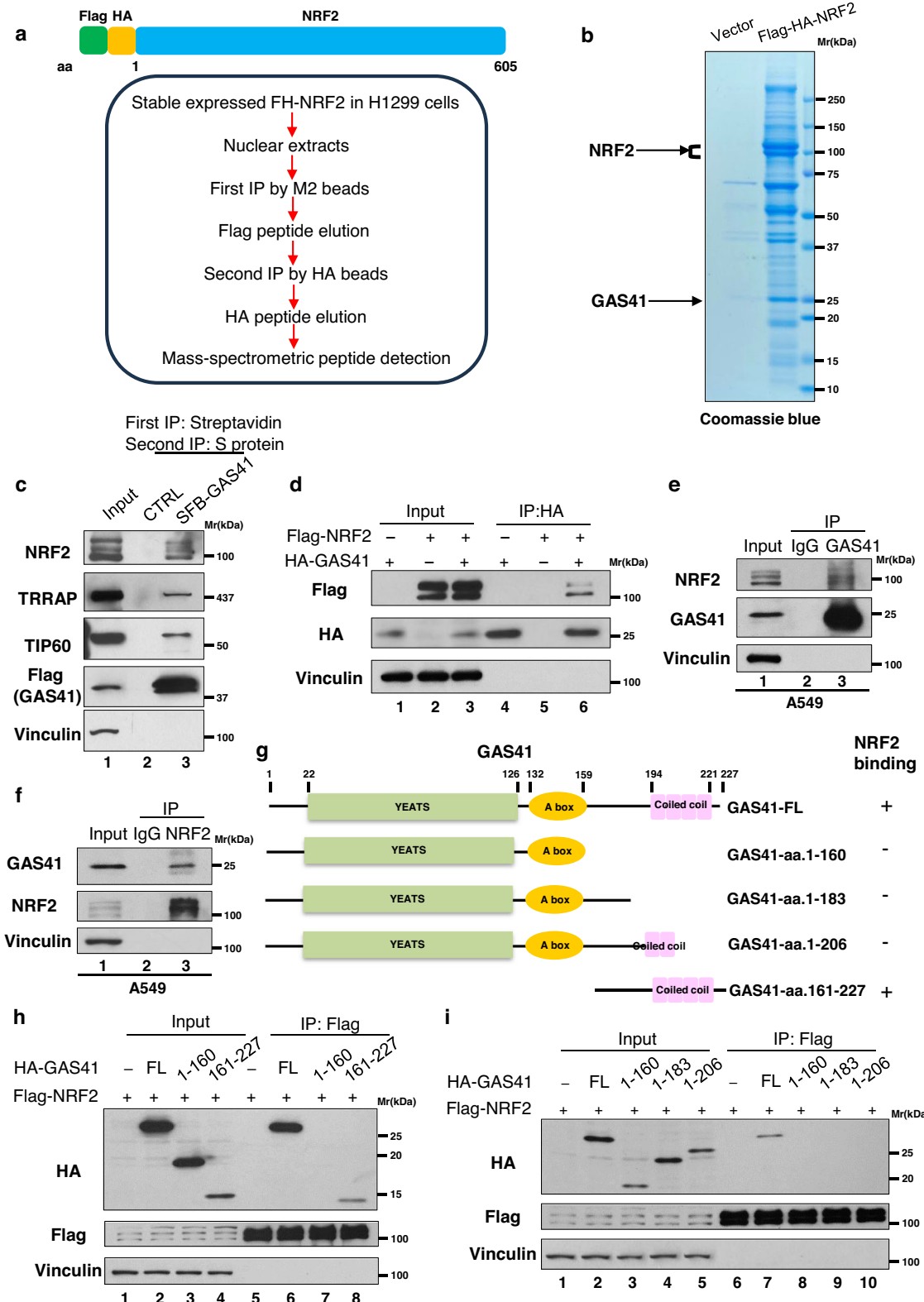

GAS41 and TRRAP or TIP60, which differed from the interaction region between GAS41 and NRF2 (Fig. 2h, i). The above data show that GAS41 interacts and regulates NRF2 transcription in a TIP60/TRRAP complex-independent manner.

GAS41 has been identified as a histone acetylation reader, which has the same character as the other two human YEATS domain-containing proteins, AF9 and ENL[13,15]. Among the multiple acetylated H3 and H4 sites, acetylation of H3 lysine 27 (H3K27-ac) is the one recognized by the GAS41 YEATS domain with the highest affinity[41]. And acetylation NRF2 is critical for its transcriptional activity[42]. Thus, we hypothesize that GAS41 may be involved in the recruitment of acetyltransferase with NRF2. To address this point, we first examined the effects of GAS41 on the interaction between well-known H3K27 acetylation acetyltransferases and NRF2. As expected, NRF2 interacted with two H3 acetyltransferases: CBP (CREB binding protein) and p300 (Supplementary Fig. 6c, d)[42]. However, in the presence of GAS41, the

**Fig. 2 | GAS41 interacts NRF2 in vitro and in vivo. a** Schematic representation of N-terminal Flag-HA-tagged NRF2 (FH-NRF2) plasmid (upper panel) and process of NRF2-interacted proteome identification (lower panel). **b** Coomassie blue staining of purified NRF2-associated protein from nuclear extraction from overexpressed FH-NRF2 H1299 stable cell line. The specific protein bands were cut and analyzed by mass spectrometry. **c** Western blot of GAS41-interacted protein complex of over-expressed SFB (S protein tag, Flag tag, and Streptavidin binding peptide)-GAS41 stable HEK293T cell line by two steps immunoprecipitations: first step IP by streptavidin beads and second step IP by S protein beads. **d** Western blot of interaction between overexpressed HA-GAS41 with Flag-NRF2 in HEK293T cells. **e** Immunoprecipitation of A549 cells by GAS41 antibody or control IgG was analyzed with western blot. **f** Immunoprecipitation of A549 cells by NRF2 antibody or control IgG was analyzed with western blot. **g** Schematic diagram of the GAS41 domains and GAS41 mutants used in this study. GAS41-amino acids 1–160 contains YEATS domain and A box (conserved sequence elements in YEATS proteins), referred to AA. 1–160; GAS41-amino acids 1–183 contains YEATS domain and A box, referred to AA. 1–183; GAS41-amino acids 1–206 contains YEATS domain, A box and partial coiled-coil motif, referred to AA. 1–206; GAS41-amino acids 161–227 contains complete coiled-coil motif, referred to AA. 161 to 227. **h** Western blot analysis of interaction between NRF2 and GAS41 mutants AA. 1–160 or AA. 161–227 in HEK293T cells. **i** Western blot analysis of interaction between NRF2 and GAS41 mutants AA. 1–160, AA. 1–183, or AA. 1–206 in HEK293T cells. Western blot experiments above (**b–f**, **h–i**) were repeated three times with similar results and representative images are shown. Source data are provided as a Source Data file.

interaction was not affected (Supplementary Fig. 6c, d). Meanwhile, we overexpressed NRF2 along with vector or GAS41 WT in A549 sgGAS41 cells and examined the CBP recruitment on the *GCLC*- and *SLC7A11*-promoter regions. As shown in Fig. 5b, overexpressed GAS41 had no obvious effects on CBP occupancy at *GCLC*- and *SLC7A11*-promoter regions. Therefore, the recruitment of acetyl-transferase is dispensable for GAS41-mediated the promotion of NRF2 transcriptional activity.

Given the interaction between GAS41 and NRF2, we investigated whether GAS41 and NRF2 exert mutual recruitment to bind to certain NRF2-target genes' promoter regions. We firstly over-expressed NRF2 with GAS41 WT, W93A, or L211A in A549 sgGAS41 cells and examined the recruitment of NRF2 to *GCLC*- and *SLC7A11*-promoter regions. As shown in Fig. 5c, NRF2 could independently bind to *GCLC*- and *SLC7A11*-promoter regions. Further, overexpressed GAS41 WT, but not GAS41 mutants, increased the occupancy of NRF2 on *GCLC* and *SLC7A11* loci, verifying that both GAS41 domains responsible for binding with NRF2 and histone modification are required for NRF2 transcriptional regulation. Moreover, NRF2 knockdown cells were transfected with GAS41 expressing plasmid along with vector or NRF2 expressing plasmids. GAS41 was observed to be enriched in the promoter region of *GCLC* and *SLC7A11* genes; however, the recruitment of GAS41 to the promoter region was significantly increased in the presence of NRF2 (Fig. 5d and Supplementary Fig. 6e), suggesting the binding of GAS41 and NRF2 on transcriptional activating chromatin contributes to the expression of antioxidant genes.

To further validate this point, CUT&RUN assays were performed by using A549 sgNC and sgGAS41 cells to analyze the effect of GAS41 on NRF2 DNA binding ability, followed by next-generation sequencing. As shown in Fig. 5e, f and Supplementary Fig. 6f, NRF2 binding activities on the promoters of *SLC7A11*, *GCLC*, and *NQO1* were significantly reduced upon loss of GAS41 expression. In line with these results, Chromatin immunoprecipitation (ChIP)-qPCR assays showed NRF2 enrichment at *SLC7A11* and *GCLC* promoter region significantly declined after losing GAS41 expression (Fig. 5g). In addition, we also found the GAS41 was not involved in other NRF2 targets' regulation, such as *ME1* (Supplementary Fig. 6g), suggesting that GAS41-mediated regulation of NRF2 is promoter-specific. Similarly, after knocking down NRF2, we found a significant decrease of GAS41 occupancy on *SLC7A11* and *GCLC* promoter region, suggesting GAS41 and NRF2 mutually recruit on the anti-oxidant genes loci (Fig. 5h and supplementary Fig. 6h). To further prove this point, *KEAP1* WT H1299 cells were treated with tBHQ, and the recruitment of GAS41 on *SLC7A11* and *GCLC* promoter was analyzed. Obviously, stabilizing NRF2 by tBHQ treatment strengthened the binding of GAS41 on *SLC7A11* and *GCLC* promoter region (Fig. 5i, j). Altogether, although GAS41 and NRF2 are both independently bound to *GCLC* and *SLC7A11* promoter region, by simultaneously interacting with both NRF2 and the H3K27-ac marker, GAS41 acts as an anchor for NRF2 on chromatin to regulate antioxidant genes expression (Fig. 5k).

## Loss of GAS41 promotes tumor suppression in vivo, at least partially through ferroptosis

Though previous study reports that GAS41 upregulation promotes NSCLC progression through cell cycle regulation[4], whether GAS41-mediated ferroptosis resistance also contributes to NSCLC tumor-igenesis is unknown. Consistent with previous reports, we found that GAS41 deficiency inhibited cell proliferation and colony formation abilities in both A549 and H460 cell lines (Supplementary Fig. 7a–d). Supplementation with antioxidant NAC (N-Acetyl-L-Cysteine), a pre-cursor of cysteine that replenishes GSH biosynthesis, could partially restore the proliferation defects caused by the GAS41 deficiency, indicating the restrained cysteine-GSH metabolism upon loss of GAS41 function (Supplementary Fig. 7b, d). To further evaluate the effects of GAS41 on tumor growth under physiological conditions, we applied a xenograft tumor model by using BALB/c nude mice and injected A549 shGAS41 Tet-On inducible cells subcutaneously. The mice were under a doxycycline-containing or doxycycline-free diet for four weeks. As shown in Supplementary Fig. 7e, f, mice with the doxycycline diet had decreased tumor growth. Meanwhile, protein levels of GAS41 declined in line with significantly decreased protein levels of SLC7A11 and GCLC, suggesting that GAS41-mediated antioxidant regulators expression is critical for tumor growth in vivo (Supplementary Fig. 7g). Immunohistochemistry (IHC) analysis of 4-hydroxynonenal (4-HNE) staining, the products of lipid peroxidation, revealed significant increases of 4-HNE strength in the GAS41 knockdown group in comparison with control groups (Supplementary Fig. 7h). Moreover, we found GAS41 knockdown substantially increased the mRNA levels of *PTGS2*, a well-known ferroptosis marker, indicating increased sus-ceptibility to ferroptosis in GAS41-deficit tumors (Supplementary Fig. 7i). To further prove this point, a H460 cells-derived xenograft tumors assay was performed. As shown in Fig. 6a, GAS41 knockdown substantially reduced the tumor weight. Meanwhile, we observed that the mRNA and protein levels of SLC7A11, GCLC, and NQO1 were sup-pressed in GAS41-deficit tumors (Fig. 6b and Supplementary Fig. 7j) accompanied by decreased GSH concentration in comparison to control tumors (Fig. 6c). We further examined the lipid peroxidation levels of tumor-derived cells by flow cytometry, which is the direct evidence for ferroptosis. As shown in Fig. 6d, the lipid peroxidation levels of GAS41-deficiency tumor cells were markedly increased in comparison to tumor cells derived from GAS41 intact tumors. The *PTGS2* mRNA levels were consistently upregulated upon GAS41 knockdown in vivo (Fig. 6e). Altogether, our data indicate that fer-roptosis is partially responsible for GAS41 knockdown-induced lung tumor suppression.

Finally, we examined whether the increased ferroptosis activity in GAS41-deficit tumors is required for tumor suppression. To this end, we tested whether the tumor growth suppression effects could be reversed in the presence of the ferroptosis inhibitor in the A549 cells-derived xenograft tumor model. BALB/c nude mice with GAS41-deficit A549 tumors were randomly divided into two groups when the tumors became palpable and intraperitoneally subjected to 1 mg/kg ferrostain-1 per day for a total of 14 days. As expected, ferrostain-1 can partially

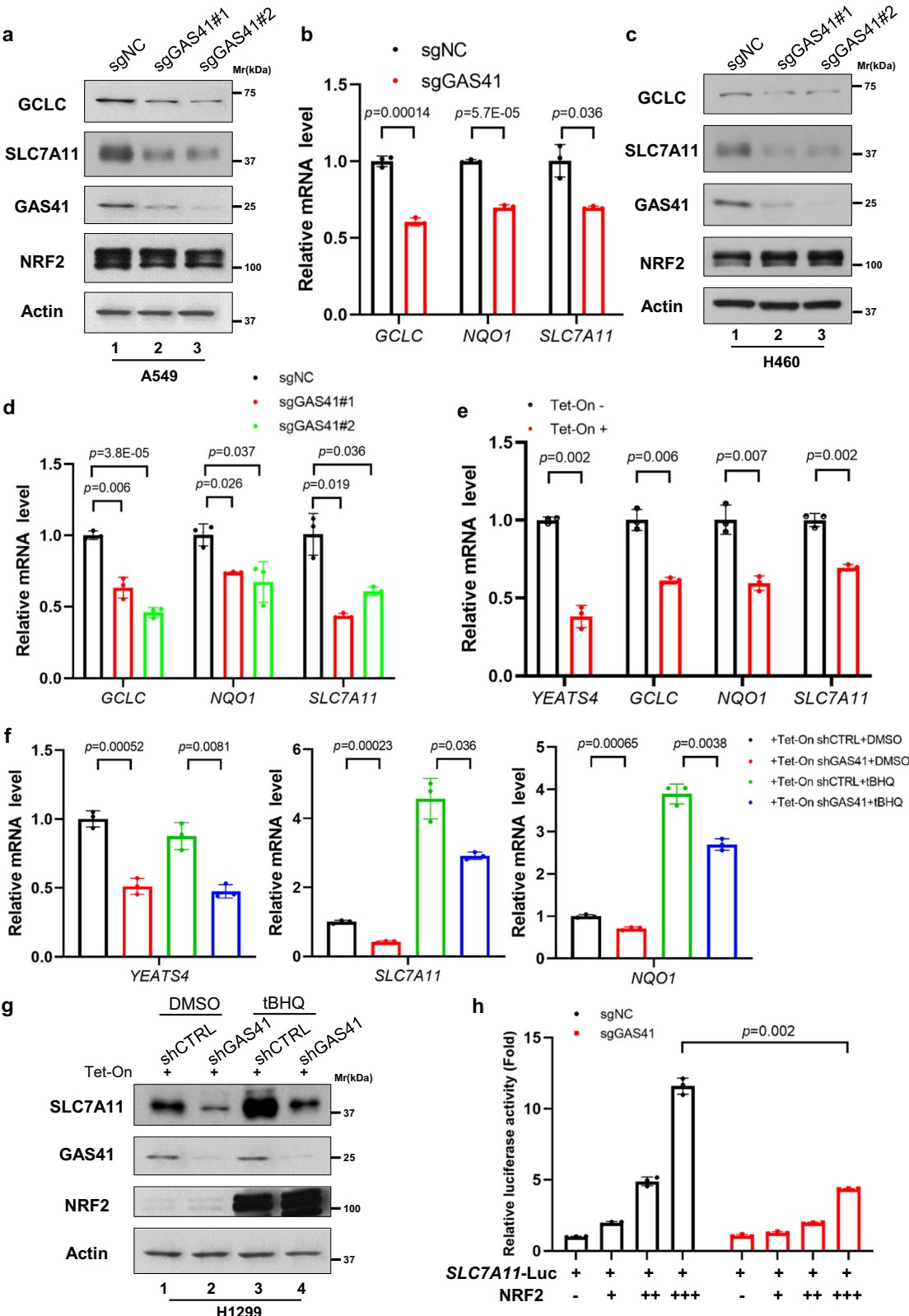

rescue GAS41 deficit-induced tumor suppression (Fig. 6f, g). Consistent with the results obtained from H460 cells-derived xenograft tumors (Fig. 6d), GAS41 knockout indeed increased the lipid peroxidation level compared with the GAS41 intact group (Fig. 6h, i); in contrast, ferrostain-1 completely eliminated the upregulation of lipid peroxidation induced by the deficiency of GAS41, indicating that ferroptotic cell death contributes to GAS41 deficit-induced tumor suppression. Thus, our data suggest that GAS41 suppresses ferroptosis in vivo and that ferroptosis is, at least, partly responsible for the overexpression of GAS41-induced tumor progression in vivo.

Clinically, it is well-reported that the cancer genetics alternations of *YEATS4* in NSCLC are most likely genomic amplification[36]. However, the transcriptional alternation upon *YEATS4* amplification is unknown. We chose LSCC for further investigation because *YEATS4* amplification

**Fig. 3 | GAS41 promotes NRF2 transcriptional ability on antioxidant genes.**
**a** Western blot of GCLC, SLC7A11, GAS41, and NRF2 protein levels of sgNC and sgGAS41 A549 cells. **b** RT-qPCR analysis of *SLC7A11*, *GCLC*, and *NQO1* mRNA levels of sgNC and sgGAS41 A549 cells. **c** Western blot of GCLC, SLC7A11, GAS41, and NRF2 protein levels of sgNC and sgGAS41 H460 cells. **d** RT-qPCR analysis of *SLC7A11*, *GCLC*, and *NQO1* mRNA levels of sgNC and sgGAS41 H460 cells. **e** RT-qPCR analysis of *YEATS4*, *SLC7A11*, *GCLC*, and *NQO1* mRNA levels of A549 shGAS41 Tet-On inducible cells incubated without or with doxycycline (0.2 µg/mL) for 72 h. **f** RT-qPCR analysis of *YEATS4*, *SLC7A11*, and *NQO1* mRNA levels in H1299 shControl (shCTRL) and shGAS41 Tet-On inducible cells incubated with doxycycline (0.2 µg/mL) for 72 h treated with tBHQ (50 µM) for 24 h. **g** Western blot of SLC7A11, GAS41,

and NRF2 protein levels of H1299 shCTRL and shGAS41 Tet-On inducible cells incubated with doxycycline (0.2 µg/mL) for 72 h treated with tBHQ (50 µM) for 24 h. **h** Loss of GAS41 reduces transcriptional activities of the *SLC7A11* promoter. The Luciferase reporter assay examined the effects of GAS41 on transcriptional activities of the *SLC7A11* promoter (*SLC7A11*-Luc) in control cells (A549 sgNC) or GAS41-null cells (sgGAS41). *Renilla* control reporter was used as a transfection internal control. Data are mean ± SD of *n* = 3 independent biological repeats. *p* values were calculated using unpaired, two-tailed Student's *t* test. Western blot experiments above (**a**, **c**, and **g**) were repeated three times with similar results and representative images are shown. Source data are provided as a Source Data file.

is the only type of genetic alteration (3%, Supplementary Fig. 7k). From the RNA profiling of LSCC, in *NFE2L2* WT and *KEAP1* WT cases, *YEATS4* genomic amplification patients has higher mRNA levels of *YEATS4* in comparison with unaltered patients suggesting that *YEATS4* amplification readily increases the expression levels (Supplementary Fig. 7l). *YEATS4* amplification samples show increased mRNA levels of several well-known antioxidant genes, including *NQO1*, *SLC7A11*, and *GCLC*, suggesting beside *NFE2L2* mutation and *KEAP1* mutation, *YEATS4* amplification could be the other critical genetic marker in lung cancer by modulating antioxidant pathways (Fig. 6j and Supplementary Fig. 7l). Moreover, by analyzing the survival data from the TCGA-LUAD dataset, the signature of high *SLC7A11*, high *GCLC*, and high *YEATS4* expression predicted even worse overall survival than either parameter alone, suggesting that SLC7A11- and GCLC-mediated GSH synthesis and ferroptosis defense is responsible for GAS41-mediated tumor progression (Fig. 6k).

## Discussion
Aberrant epigenetic reprogramming has been reported to play an important role in cancer progression[43]. Previous study indicates that GAS41 deficiency suppresses NSCLC cell growth and survival[4]. Nevertheless, the precise mechanism by which GAS41 contributes to tumor development needs further elucidation. Here, we discovered GAS41 as a critical modulator in regulating cystine-GSH metabolism to promote tumor growth at least partially through repressing ferroptosis. Our study revealed an unexpected epigenetic regulation of GAS41-mediated regulation in transcription by anchoring NRF2 on chromatin (Fig. 5k).

YEATS4 amplification is detected in 3% of LSCC and 5% of LUAD from TCGA-pan cancer database (Supplementary Fig. 7k)[36] and GAS41 overexpression is also implicated in NSCLC[4]. Patients with higher *YEATS4* expression levels have significantly shorter overall survival compared to lower expression levels of *YEATS4* in LUAD (Supplementary Fig. 7m). Therefore, GAS41 has emerged as an oncogene and target in NSCLC. We found that loss of GAS41 sensitizes the cancer cells to ferroptosis through downregulating specific GSH synthesis pathway-related gene expression. Consequently, GAS41 knockdown led to tumor suppression in NSCLC, at least partially, through ferroptosis. It is interesting that the rewiring of a metabolic pathway by *YEATS4* amplification in NSCLC facilitates the intrinsic response to oxidative stress and further promotes cancer progression. Thus, *SLC7A11* and *GCLC* expression could be defined as a cancer signature of GAS41 amplification in NSCLC (*KEAP1* and *NFE2L2* WT). Previous studies indicate that GAS41 plays important roles in modulating other growth suppression pathways including apoptosis[5,8,44]. Future investigations are required to test whether inhibition of both ferroptosis and apoptosis is able to completely rescue the growth suppression effects induced by loss of GAS41. Notably, recent studies have reported that small molecule inhibitors of GAS41 binding to the lysine acetylation recognition site of GAS41 were discovered[45,46], providing an effective tool to have a better understanding of the GAS41 role in NSCLC. Since our studies demonstrate that GAS41-mediated histone binding is critical for its effects on NRF2-dependent transactivation, it

will be interesting to examine whether the combination of those small molecule inhibitors of GAS41 (or loss of GAS41) and ferroptosis inducers may have synergistic effects in tumor suppression for potential cancer therapy.

GAS41 recognizes histone acetylation through the YEATS domain, which facilitates H2A.Z deposition catalyzed by chromatin remodelers and thereby regulates DNA replication and cell cycle gene expression[4]. However, GAS41 regulated specific GSH metabolism gene expression through interacting with NRF2, suggesting that GAS41 functions as an oncogene through a distinct transcriptional mechanism. Our study broadens the scope of GAS41-required transcriptional mechanisms and has a profound understanding of GAS41 oncogenic function.

## Methods
### Mice
All the mice were housed in a temperature-controlled room (65–75 °F) with 40–60% humidity, with a light/dark cycle of 12/12 h. All animal experiments were conducted with the approval and complied with all relevant ethical regulations of the Institutional Animal Care and Use Committee (IACUC) at Columbia University under the supervision of the Institute of Comparative Medicine (ICM). Six-week-old female nude mice (Nu/Nu, Charles River, RRID: IMSR_CRL:088) were used for xenograft experiments.

### Cell culture
Human embryonic kidney 293T (CRL-3216), human melanoma cell A375 (*p53* WT) (CRL-1619), human lung large cell carcinoma H460 (*p53* WT; *KEAP1* mutant) (HTB-177), human lung adenocarcinoma (LUAD) A549 (*p53* WT; *KEAP1* mutant) (CCL-185), and human non-small cell lung carcinoma H1299 (*p53* null; *KEAP1* WT) (CRL-5803) cells were previously obtained from ATCC. Cells were cultured with DMEM supplemented with 10% (v/v) FBS (Gibco), 100 units/ml penicillin, and 100 µg/ml streptomycin (Gibco) in a humidified incubator at 37 °C with 5% $CO_2$. Two-tenths microlitre of doxycycline was used in Tet-On inducible cells. All cell lines have been regularly tested to be negative for mycoplasma contamination every month. No cell lines used in this work were listed in the International Cell Line Authentication Committee database. The cell lines were freshly thawed from the purchased seed cells and were cultured for no more than one month. The morphology of cell lines was checked before the experiments and compared with the ATCC cell line image to avoid cross-contamination or misuse of cell lines.

### CRISPR-Cas9 knockout screen, high-throughput sequencing, and analysis
Liu human CRISPR-Cas9 knockout library (H1 and H2) was a gift from Xiaole Shirley Liu (Addgene #1000000132). The detailed processes of CRISPR-Cas9 screens and bioinformatics analysis have been described previously[47]. Briefly, $1.25 \times 10^8$ A375 cells (around 400X coverage) were infected with the H2 library at a multiplicity of infection (MOI) of around 0.3. 48 h after transduction, $T_0$ cells were harvested. The rest of the cells were selected with 0.9 µg/ml

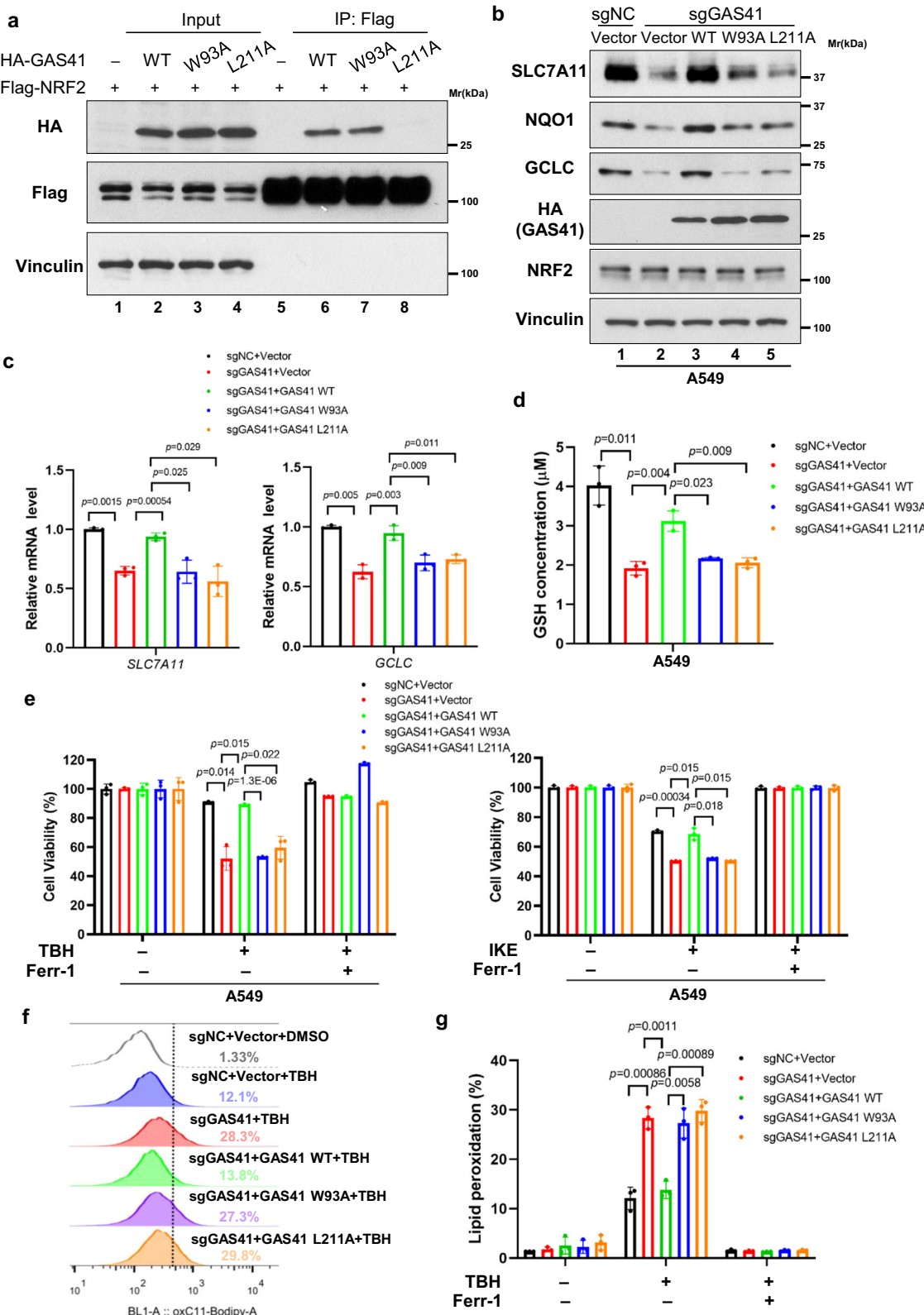

puromycin for 48 h, and $5 \times 10^7$ cells were passed every 48–72 h for 14 days. Then, $T_{14}$ cells were harvested. For selection, $1 \times 10^8$ cells were treated with TBH (300 μM) and Nutlin-3 (5 μM) for 12 h. After the indicated treatment, live cells were collected, and genomic DNA was isolated using the Wizard Genomic DNA Purification Kit (Promega). The high-throughput sequencing libraries were prepared through two-step PCR amplification. Seventy-five nucleotides reads were done using the Illumina HiSeq2500 at the Columbia University Genome Center. MAGeCK (version 0.5.7) software was used for screen analysis.

**Plasmid generation, siRNA, and transfection**
Human GAS41 full-length cDNA was amplified from A549 cells cDNA reversed by SuperScript IV VILO Master Mix (Invitrogen).

**Fig. 4 | The interaction between GAS41 and NRF2 is required for NRF2 transcriptional ability. a** Western blot analysis of interaction between NRF2 and GAS41 mutants (W93A and L211A) in HEK293T cells. **b** Western blot of SLC7A11, GCLC, and NQO1 protein levels of A549 sgNC re-expressed with vector, sgGAS41 cells re-expressed with vector, GAS41 WT, GAS41 W93A mutant or GAS41 L211A mutant. **c** RT-qPCR analysis of *SLC7A11* and *GCLC* mRNA levels of A549 sgNC re-expressed with vector, sgGAS41 cells re-expressed with vector, GAS41 WT, GAS41 W93A mutant or GAS41 L211A mutant. **d** Measurement of GSH concentration of A549 sgNC re-expressed with vector, sgGAS41 cells re-expressed with vector, GAS41 WT, GAS41 W93A mutant or GAS41 L211A mutant. **e** Cell viability of

A549 sgNC re-expressed with vector, sgGAS41 cells re-expressed with vector, GAS41 WT, GAS41 W93A mutant, or GAS41 L211A mutants treated with TBH (120 μM, left panel) for 4 h or IKE (3 μM, right panel) for 24 h. **f, g** Assessment of lipid peroxidation (**f**) and statistical bar graph (**g**) by flow cytometry after C11-BODIPY staining of A549 sgNC re-expressed with vector, sgGAS41 cells re-expressed with vector, GAS41 WT, GAS41 W93A mutant, or GAS41 L211A mutant. Data are mean ± SD of $n = 3$ independent biological repeats. $p$ values were calculated using unpaired, two-tailed Student's $t$ test. Western blot experiments above (**a**, **b**) were repeated three times with similar results, and representative images are shown. Source data are provided as a Source Data file.

GAS41 fragment mutants were sub-cloned into pRK5 or pMH-SFB vectors. Point mutants of GAS41 plasmids were constructed by using the QuickChange XL Site-Directed Mutagenesis Kit (Agilent) according to the standard protocol. NRF2 cDNA generated as described[38] was sub-cloned into pRK5. CbS-Flag-TRRAP was a gift from Michael Cole & Yardena Samuels (Addgene plasmid #32103; http://n2t.net/addgene:32103; RRID: Addgene_32103). CMV-TIP60 and pcDNA3.1/v5-His-Topo-SLC7A11 expressing plasmid was generated as previously[48,49]. pCDNA3 NQO1 was a gift from Yosef Shaul (Addgene plasmid # 61730; http://n2t.net/addgene:61730; RRID: Addgene_61730). *GCLC* cDNA was purchased from Dharmacon (MHS6278-202759380) and sub-cloned into pLVX-M-puro, which was a gift from Boyi Gan (Addgene plasmid #125839). Transfection of expressing plasmids was conducted by Lipofectamine 3000 Reagent (Invitrogen, L3000150) according to the manufacturer's protocol.

siRNA transfection was performed using Lipofectamine 3000 Reagent for 24 h and then transfected again according to the manufacturer's instructions. TIP60 siRNA was generated as previously[48]. TRRAP siRNA was purchased from Dharmacon Reagents (L-005394-00).

### Reagents and compounds
Nutlin-3 (N6287), tert-butyl hydroperoxide (TBH, 458139), N-acetyl-L-cysteine (NAC, A7250), tert-butylhydroquinone (tBHQ, 112941), 1S,3R-RSL-3 (RSL-3, SML2234), and ferrostatin-1 (Ferr-1, SML0583) were obtained from Sigma-Aldrich. Imidazole ketone erastin (IKE, HY-114481) was obtained from MedChemExpress.

### Cystine starvation treatment
For cystine starvation medium, glutamine, methionine, and cystine-deficient DMEM (21013024, Invitrogen) was supplemented with 4 mM glutamine (Sigma-Aldrich, G8540), 200 μM methionine (Sigma-Aldrich, M5308), and 10% FBS. For cystine (Sigma-Aldrich, C7602) containing medium, 200 μM cystine was added back to the cystine starvation medium. Before seeding, cells were pre-washed with 1× PBS three times and then split into cystine starvation medium or cystine-containing medium for the indicated time.

### Cell viability assay
Unless otherwise specified, cells were seeded in white, sterile, and tissue culture-treated opaque 96-well microplate (PerkinElmer) at $5 \times 10^3$ cells per well. About 18 h after cell seeding, cells were treated with indicated compounds at the indicated concentrations for the indicated time. There were three biological replicates per condition. Cellular ATP levels were quantified using CellTiter-Glo 2.0 reagent (Promega) following the manufacturer's instructions by GloMax Discover Microplate Reader. Relative cell viability was measured in comparison to the relative untreated condition. Nonlinear regression analysis of the mean ± SD $n = 3$ biological replicates of each data point was used to measure the fit curves of cell viability by GraphPad Prism 8.0. To calculate the cell death ratio by this method, the percentage of cell death was counted as 100 minus the percentage of cell viability.

### Cell death assay
Cells were seeded in 12-well plate (Corning) at $4 \times 10^4$ cells per well. About 18 h after cell seeding, cells were pre-treated with indicated compounds for the indicated time before further treatment or directly treated with indicated compounds at the indicated concentrations for the indicated time. 30 nM SYTOX green dead cell stain (Invitrogen, S34860) was added into plates and incubated for 1 h at 37 °C. At least three randomly chosen bright fields and fluorescence fields were captured by microscopy (Olympus, IX51). Living cells (without green stained) and dead cells (with green stained) were counted, and the cell death ratio was calculated by the number of dead cells/numbers of (living cells + dead cells).

### GSH detection
For adherent cells, A549 sgNC and sgGAS41 cells were transfected with indicated plasmids for 48 h before this experiment and re-seeded in white, sterile, and tissue culture treated opaque 96-well microplate (PerkinElmer) at $5 \times 10^3$ cells per well with three biological replicates per group. Eighteen hours after seeding, GSH concentration was measured by following the manufacturer's instructions of the GSH/GSSG-Glo Assay (V6611, Promega) through GloMax Discover Microplate Reader.

For tumor tissue, isolated tumor tissue from mice was perfused with 1× PBS containing heparin to remove blood and clots. Ten milligram tumor tissue was homogenized in 1 mL 1×PBS containing 2 mM EDTA and centrifuged to collect the supernatant. GSH concentration was measured by following the manufacturer's instructions of the GSH/GSSG-Glo Assay through the GloMax Discover Microplate Reader.

### Co-immunoprecipitation (IP) assay
Cells were harvested and rinsed with 1× PBS twice. Cells were lysed with BC150 Buffer (50 mM Tris-HCl pH 7.3, 150 mM NaCl, 0.1 mM EDTA, 0.4% NP-40, and 10% glycerol) added protease inhibitor cocktail, 1 mM dithiothreitol (DTT), and 1 mM phenylmethyl sulfonyl fluoride (PMSF) on ice for 1 h, following sonication for 20 s. Cell lysates were centrifuged for 15 min at 20000× g, and the supernatant was collected. Then, the same total protein quantified with Protein Assay Dye Reagent (Bio-Rad, 5000006) was taken for IP assay. Two microgram of the indicated antibody was added into lysates and incubated overnight at 4 °C, followed by the addition of 20 μL protein A/G agarose for 2 h. For commercial conjugated beads, lysates were added 20 μL Flag M2 Affinity Gel (Sigma, A2220), HA agarose (Sigma-Aldrich, A2095), or Streptavidin agarose (Sigma-Aldrich, 16–126) and incubated overnight at 4 °C. Next day, beads were washed with BC200 buffer (same formula as BC150 besides 200 mM NaCl) two times and BC150 buffer three times, and the complex was eluted by 1× Loading buffer, 0.1 M glycine (pH = 2.6), or Biotin for further western blot analysis.

### Protein purification
HEK293T cells were transfected with Flag-tagged GAS41 plasmids. Forty-eight hours after transfection, cells were harvested and lysed in BC500 (same formula as BC150 besides 500 mM NaCl), following sonication for 3 min. An appropriate amount of Flag M2 Affinity Gel (Sigma-Aldrich, A2220) was added to the supernatant and incubated

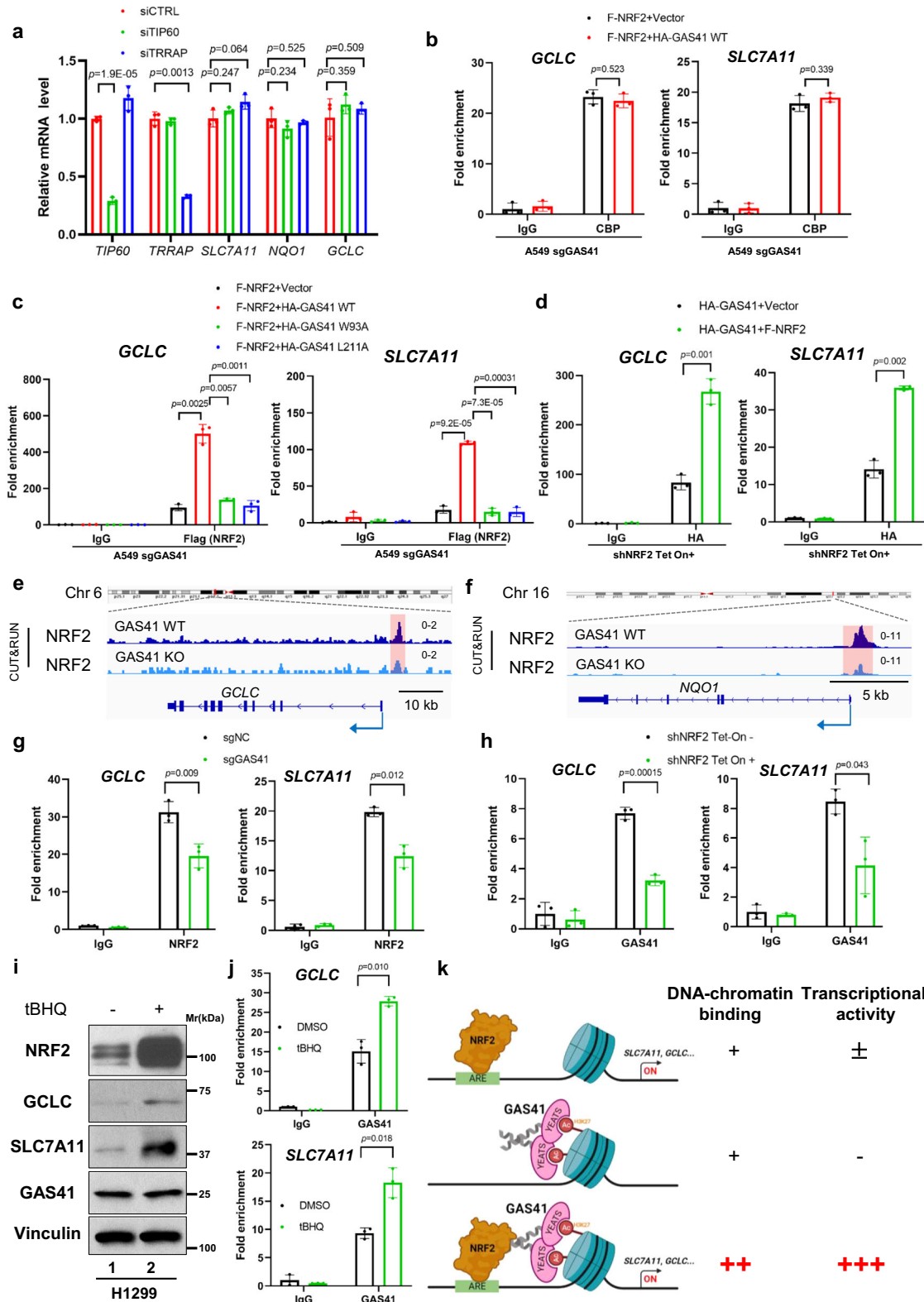

overnight at 4 °C. The next day, beads were rinsed with BC500 buffer six times, and the purified proteins were eluted with Flag peptide diluted in BC20 buffer (50 mM Tris-HCl pH 7.3, 20 mM NaCl, 0.1 mM EDTA, and 10% glycerol) several times. Finally, a certain amount of protein was measured by SDS-PAGE gel and stained using GelCode Blue Stain Reagent (ThermoFisher Scientific, 24592). Bovine serum albumin proteins (BSA) were used for quantification.

**In vitro GST pull-down assay**

NRF2 full-length or fragments were generated as described previously. GST, GST-NRF2-NT, GST-NRF2-M, or GST-NRF2-CT proteins were inducibly expressed in *Rosetta* bacterial cells and incubated with GST Bind Resin (Novagen, 70541). Equal amounts of purified GAS41 proteins were incubated with equal amounts of corresponding GST-tagged proteins in BC200 buffer for 4 h at 4 °C, followed by washing

**Fig. 5 | The regulatory interplay of NRF2 and GAS41 in transcriptional regulation. a** RT-qPCR analysis of *TIP60*, *TRRAP*, *SLC7A11*, *NQO1*, and *GCLC* mRNA levels in A549 cells transfected with control siRNA (siCTRL), TIP60 siRNA (siTIP60), or TRRAP siRNA (siTRRAP). **b** ChIP-qPCR analysis of CBP binding on *SLC7A11* and *GCLC* promoter region in A549 sgGAS41 cells transfected with NRF2 along with vector or GAS41 WT. **c** ChIP-qPCR analysis of overexpressed NRF2 binding on *SLC7A11* and *GCLC* promoter region in A549 sgGAS41 cells transfected with NRF2 expressing plasmid along with vector, GAS41 WT, GAS41 W93A mutant, or GAS41 L211A mutant. **d** ChIP-qPCR analysis of overexpressed GAS41 enrichment on *GCLC* and *SLC7A11* promoter region in A549 shNRF2 Tet-On inducible cells pre-incubated without or with doxycycline (0.2 µg/mL) for 48 h, then transfected with HA-tagged GAS41 expressing plasmid along with vector or Flag-tagged NRF2 expressing plasmid. **e, f** Snapshot of NRF2 CUT&RUN signal in A549 sgNC and sgGAS41 cells at *GCLC* (**e**) and *NQO1* (**f**) genes loci. **g** ChIP-qPCR analysis of NRF2 binding on *SLC7A11* and *GCLC* promoter region in A549 sgNC and sgGAS41 cells. **h** ChIP-qPCR analysis of GAS41 enrichment on *SLC7A11* and *GCLC* promoter region in shNRF2 A549 without or with doxycycline (0.2 µg/mL) for 72 h. **i** Western blot of GAS41, NRF2, SLC7A11, and GCLC protein levels in H1299 treated with DMSO or tBHQ (50 µM) for 24 h. **j** ChIP-qPCR analysis of GAS41 enrichment at *SLC7A11* and *GCLC* promoter region in H1299 treated with DMSO or tBHQ (50 µM) for 24 h. **k** Work model for the role of GAS41 in antioxidant transcription regulation through anchoring NRF2 with histone acetylation. Created with BioRender.com. Data are mean ± SD of *n* = 3 independent biological repeats. *p* values were calculated using unpaired, two-tailed Student's *t* test. The experiment (**i**) was repeated three with similar results and representative results are shown. Source data are provided as a Source Data file.

with BC200 buffer six times. The binding complex was eluted by boiling with 1× Loading buffer for western blot analysis.

### Antibodies and immunoblotting

Cells were lysed with Flag lysis buffer [50 mM Tris-HCl (pH 7.9), 137 mM NaCl, 1% Triton X-100, 0.2% Sarkosyl, 1 mM NaF, 1 mM $Na_3VO_4$, and 10% glycerol] containing protease inhibitor cocktail and 1 mM DTT, and 1 mM PMSF. The same amount of protein from different experiment groups was quantified and detected with western blot analysis. Western blot was conducted for protein analysis according to standard methods with 4–20% pre-cast SDS-PAGE gel (Invitrogen, XP0420). Commercial antibodies are shown as follows: Flag (Sigma-Aldrich, F3165, 1:5000 dilution), Vinculin (Sigma-Aldrich, V9131, 1:10000 dilution), Actin (Sigma-Aldrich, A3853, 1:5000 dilution), HA (Roche, 11867423001, 1:2000 dilution), NRF2 (Abcam, ab62352, 1:200 dilution), GPX4 (Abcam, ab125066, 1:1000 dilution), NRF2 (Cell signaling technology, 12721, 1:200 dilution), SLC7A11 (Cell signaling technology, 12691, 1:1000 dilution), p53 (DO-1) (Santa Cruz, sc-126, 1:10000 dilution), GAS41 (Santa Cruz, sc-393708, 1:200 dilution), FSP1 (AMID) (Santa Cruz Biotechnology, sc-377120, 1:1000 dilution), GCLC (proteintech, 12601-1-AP, 1:10000 dilution), NQO1 (proteintech, 11451-1-AP, 1:10000 dilution), TIP60 (proteintech, 10827-1-AP, 1:500 dilution), and DHODH (Proteintech, 14877-1-AP, 1;10000 dilution). Peroxidase AffiniPure Goat Anti-Mouse IgG (H + L) (Jackson Immunoresearch, 115-035-146, 1:5000 dilution), Peroxidase AffiniPur Goat Anti-Rabbit IgG (H + L) (Jackson Immunoresearch, 111-035-045, 1:5000 dilution), and Goat Anti-Rat IgG(H + L) (SouthernBiotech, 3050-05, 1:5000 dilution) were detected by ECL (Thermo Fisher Scientific, 32106).

### Real-time quantitative PCR (RT-qPCR)

Total RNA of A549, H460, and H1299 cells or tumor tissues were isolated using TRIzol reagent (Invitrogen) according to the manufacturer's instructions. cDNA was reversed by SuperScript IV VILO Master Mix (Invitrogen). qPCR was then performed using SYBR Green Master Mix (Invitrogen) to detect the mRNA expression levels of indicated genes with Applied Biosystems 7500 Fast Dx Real-Time PCR Instrument. The expression levels of target genes were normalized by *ACTB*. The following primers are shown in Supplementary Table 1.

### CUT&RUN and library preparation

CUT&RUN assay was performed using the ChIC/CUT&RUN Assay Kit (Active&Motif, 53180) with minor modifications. Briefly, 0.5 million A549 NC or sgGAS41 cells were bound to Concanavalin A conjugated paramagnetic beads (EpiCyher) and then incubated with 1 µg NRF2 antibodies overnight. The next day, the cells were incubated with pAG-MNase at 25 °C for 10 min. After pAG-MNase binding, the reaction was extensively washed. Digestion of chromatin was initiated by the addition of 2 mM $CaCl_2$ at 4 °C for 60 min. The DNA fragments were released from the cells by adding the Stop Buffer and then extracted by QIquick PCR Purification Kit (QIAGEN, 28104). The xGEN ssDNA&Low-Input DNA Library Prep Kit (Integrated DNA Technologies) was used for preparing libraries. The libraries were pooled and sequenced using paired-end sequencing on Illumina NextSeq 500 platforms at the Columbia University Genome Center.

### Analysis of CUT&RUN

Raw reads were trimmed to remove sequencing adapters and low-quality reads were removed using Trim Galore (version 0.6.7) with default parameters.

Sequence reads were mapped back to the human (hg38) reference genome using Bowtie2 (version 2.2.4) with –no-mixed –no-discordant –no-dovetail –no-contain –local parameters. Coverage for each base pair of the human genome was computed using genomeCoverageBed from BEDTools (version 2.29.2) and normalized to library size (reads-per-million; RPM). Sparse Enrichment Analysis for CUT&RUN (SEACR version 1.3) was used for peak calling. Overlapped peaks were extracted using BEDTools (version 2.29.2). Bigwig files were generated using UCSC bedGraphToBigWig (version 359); scores represent the normalization reads density. The gene tracks were visualized by IGV(version 2.17.0).

### Chromatin immunoprecipitation (ChIP)-qPCR

Indicated cells were fixed with 1% formaldehyde for 15 min. Medium was added 0.125 M glycine for 10 min incubation to stop fixing. Then, cells were harvested and lysed with ChIP lysis buffer (10 mM Tris-HCl pH 8.0, 5 mM EDTA,150 mM NaCl, and 0.5% NP-40) containing a protease inhibitor cocktail for 10 min on ice. After that, the cell pellet was collected and sonicated with RIPA lysis buffer (10 mM Tris-HCl pH 8.0, 5 mM EDTA,150 mM NaCl, 0.1% SDS, 1% Triton X-100, and 0.5% DOC) containing protease inhibitor cocktail. The cell lysate was pre-cleaned with Protein A salmon sperm DNA agarose (Millipore, 16–157) for 2 h at 4 °C. Equal protein amount of cell lysate was incubated with 2 µg GAS41 (Sanra Cruz, sc-393708), NRF2 (Abcam, ab62352), Flag M2 (Sigma-Aldrich, F-3165), CBP (Santa Cruz, sc-369), or HA (Roche, 11867423001) antibodies or 2 µg corresponded mouse, rabbit, or rat IgG control overnight at 4 °C. Next day, 20 µl Protein A salmon sperm DNA agarose was coupled with antibody for 4 h at 4 °C. and then rinsed with RIPA lysis buffer, High Salt Wash buffer (20 mM Tris-HCl pH 8.0, 500 mM NaCl, 5 mM EDTA, 0.1% SDS, 1% Triton X-100), LiCl Wash Buffer (10 mM Tris-HCl pH 8.0, 1 mM EDTA, 250 mM LiCl, 1% DOC, 1% NP-40), and TE buffer (10 mM Tris-HCl pH 8.0, 1 mM EDTA) for once, respectively. Coupled DNA-protein complex was eluted in Elution buffer (1% SDS and 100 mM $NaHCO_3$) twice and cross-linking was reversed at 65 °C overnight incubation. The binding DNA was extracted by using QIquick PCR Purification Kit. ChIP samples were quantified by qPCR. DNA immunoprecipitated by IgG control served as a negative control. Primers used for ChIP-qPCR were listed in Supplementary Table 1.

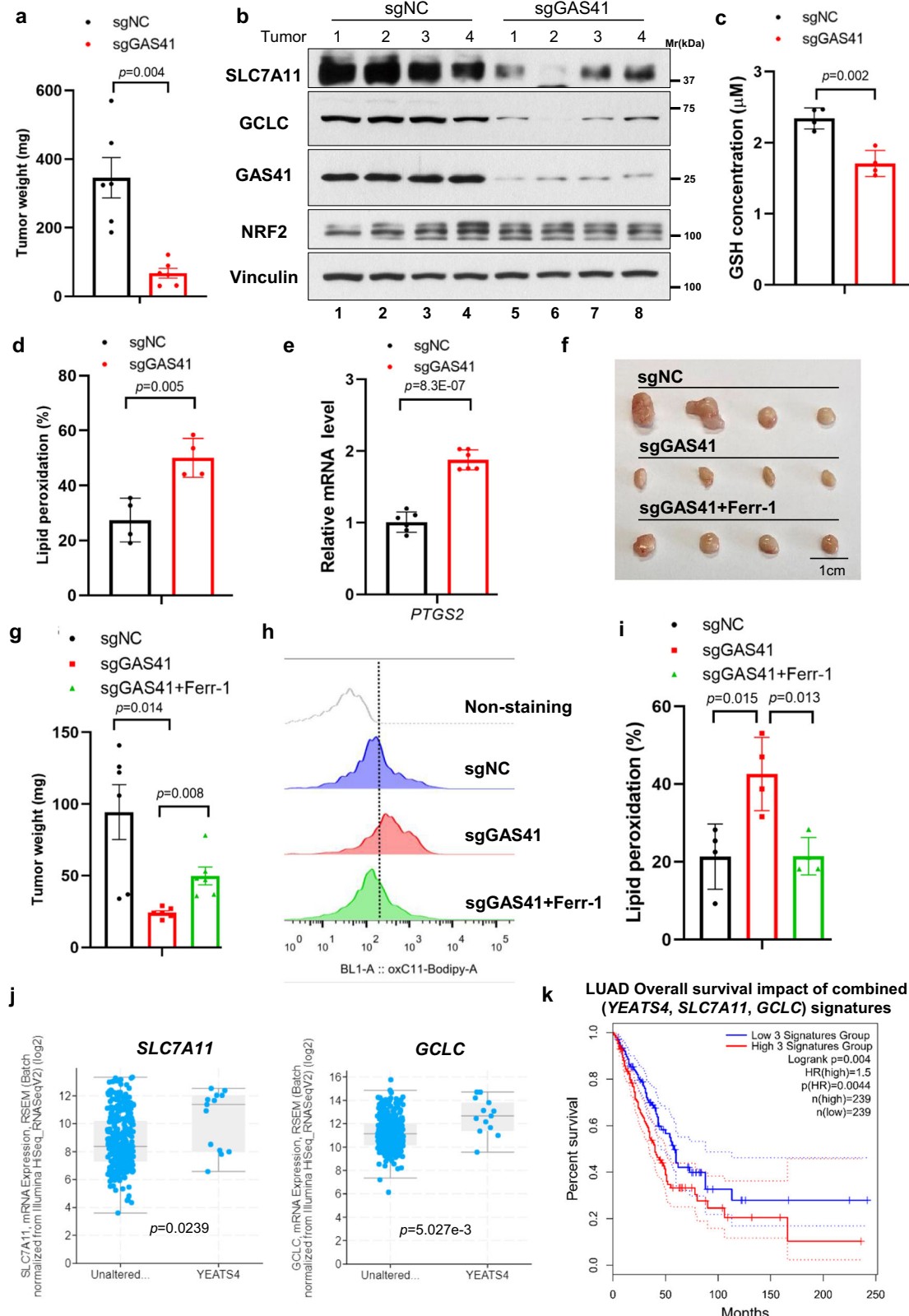

## CRISPR-Cas9 system-mediated gene ablation

p53 knockout cells were previously described[26]. To produce GAS41 sgRNAs expressing cell lines, indicated cells were co-transfected gene TrueGuide Synthetic guide RNA (Invitrogen, A35533) or Negative Control non-targeting RNA (Invitrogen, A35526) with TrueCut Cas9 protein v2 (Invitrogen, A36498) by Lipofectamine CRISPRMAX Transfection Reagent (Invitrogen, CMAX00015). After 48 h transfection,

parts of pooled cells were seeded for cell viability or western blot validation, and the rest of the cells were split into single cells for colony culture and further analysis. GAS41 sgRNAs were designed by CRISPOR and synthesized by Synthego. The sgRNA sequences are shown below:

*YEATS4*#1: TTTACTCTCCCGCCGGAGTC
*YEATS4*#2: TAGTTTACGGTAATGTTGCT

Fig. 6 | **Loss of GAS41 promotes tumor suppression, at least partially through ferroptosis in vivo. a** Weight quantification of tumors derived from sgNC and sgGAS41 H460 cells as indicated. **b** Western blot of GAS41, SLC7A11, GCLC, and NRF2 protein levels of tumor tissues in (**a**). **c** Measurement of GSH concentration of tumor tissues in (**a**). **d** Assessment of lipid peroxidation by flow cytometry after C11-BODIPY staining of tumor tissues in (**a**). **e** RT-qPCR analysis of *PTGS2* mRNA expression levels of tumor tissues in (**a**). **f, g** Image (**f**) and weight quantification (**g**) of xenograft tumors derived from sgNC, sgGAS41 A549, or sgGAS41 A549 cells administrated with Ferr-1 as indicated. **h, i** Assessment of lipid peroxidation (**h**) and statistical bar graph (**i**) by flow cytometry after C11-BODIPY staining of tumor tissues in (**f**). **j** Box plot of *SLC7A11* and *GCLC* mRNA expression of *YEATS4*-unaltered (*NFE2L2, KEAP1, YEATS4* WT, *n* = 328) or -amplified (*n* = 13) patients from TCGA-LSCC (lung squamous cell carcinoma) analyzed by cBioPortal. The median value is shown in the box. The whiskers indicate the value of minima and maxima, and the box bounds indicate the value of the first quartile or third quartile. **k** Kaplan–Meier plots of TCGA-LUAD (lung adenocarcinoma) patients stratified by unsupervised clustering on *YEATS4, SLC7A11*, and *GCLC* expression. The blue line has lower *SLC7A11*, lower *GCLC*, and lower *YEATS4* expression (*n* = 239), while the red line has higher *SLC7A11*, higher *GCLC*, and higher *YEATS4* expression (*n* = 239). For **c, d**, and **i** data are mean ± SD of *n* = 4, for **e** data are mean ± SD of *n* = 6 independent biological repeats. For **a, g** data are mean ± SEM of *n* = 6 independent tumor samples. *p* values were calculated using unpaired, two-tailed Student's *t* test. Source data are provided as a Source Data file.

## Tet-On system-mediated knocking down GAS41 or NRF2 expression

TRIPZ-inducible lentiviral GAS41 and NRF2 shRNAs were obtained from Dharmacon (RHS4696-200682959 for shGAS41#1, RHS4696-200685680 for shGAS41#2, V3THS_306092 for shNRF2-#1 and V3THS_306096 for shNRF2-#2). TRIPZ-inducible lentiviral non-silencing shRNA control was sub-cloned with the sequence which has minimal homology to known mammalian genes (ACCTCCACCCT-CACTCTGCCAT). HEK293T cells were co-transfected with Inducible lentiviral plasmids and viral packaging plasmids (psPAX2 and pMD2G) according to the radio of 4:3:1 by Lipofectamine 3000 Reagent. Forty-eight hours after transfection, medium incubated with HEK293T cells was collected and strained with 0.45 μm PES filters. Next, A549 or H1299 cells were infected with a filtered virus-containing medium with 8 μg/mL of polybrene (Santa Cruz Biotechnology). Transduced A549 cells were diluted in 10 cm dish for selection of single clones and selected with 1 μg/mL puromycin for 2 weeks. The efficiency of GAS41 or NRF2 protein knockdown was validated by western blot.

## Dual-luciferase assay

Luciferase reporter containing *SLC7A11* promoter sequences have been described previously[38]. Briefly, *SLC7A11* reporter plasmids, *Renilla* control reporter plasmids, and NRF2 expression vectors were co-transfected into A549 sgNC or sgGAS41 cells in 12-well plates. After 24 h, the relative luciferase activity was measured according to the manufacturer's protocol of the Dual-Luciferase Reporter Assay System (Promega, E1910) by GloMax Discover Microplate Reader (Promega).

## Cell proliferation and colonic formation assay

For cell proliferation assay, indicated cells were seeded at $2 \times 10^4$ cells per well in 6-well plates and incubated at indicated culture conditions for an overall 6 days. For every 48 h, total cells were digested by trypsin, collected, and counted with a hemocytometer using the standard protocol.

For the cell colonic formation assay, indicated cells were seeded at $1 \times 10^4$ cells per well in 6 cm plates and incubated at indicated culture conditions for 10 days. Cells were washed with 1× PBS twice, fixed with 4% paraformaldehyde for 20 min, stained with 0.2% crystal violet solution for 20 min, and photographed using a digital scanner.

## Lipid peroxidation analysis using C11-BODIPY

Cells were seeded in a 6-well plate at $12 \times 10^4$ cells per well. About 18 h after cell seeding, cells were pre-treated with indicated compounds for the indicated time before further treatment or directly treated with indicated compounds at the indicated concentrations for the indicated time. Cells were incubated with 2 μM BODIPY™ 581/591 C11 dye (Thermo Fisher Scientific, D3861) for 30 min at 37 °C. Then cells were harvested, washed by 1× PBS twice, and resuspended in 500 μL 1× PBS, followed through a 35 μm cell strainer (Falcon, 352235) for flow cytometry analysis. Lipid peroxidation levels were measured with an Attune NxT Acoustic Focusing Cytometer (Thermo Fisher Scientific) through the BL1 channel by analyzing 10,000 cells.

For Xenografts-derived cells, isolated tumor tissues were cut into small enough pieces and digested with collagenase type I (Thermo Fisher Scientific, 17100017) at 37 °C incubator for 1 h. The contents were passed a 35 μm cell strainer, rinsed with 1× PBS twice, and resuspended with 1× PBS. Finally, cells were stained with 2 μM BOD-IPY™ 581/591 C11 dye for 25 min at 37 °C in the dark. Then, the stained cells were rinsed by 1x PBS twice and resuspended in 1× PBS for flow cytometry analysis. Lipid peroxidation levels were measured with an Attune NxT Acoustic Focusing Cytometer through the BL1 channel by analyzing 10,000 cells.

## Immunohistochemical (IHC) staining

Tumor samples were fixed in 10% formalin for 24 h and 70% ethanol before being subjected to standard dehydration processing for preparing the paraffin blocks. Paraffin blocks were sectioned at 4 μM thickness for IHC staining. Tissue sections were deparaffinized with xylene and followed with gradient ethanol (100%, 95%, 90%, 80%, and 70%). After rinsing with deionized water for 5 min, the sections were incubated in 3% hydrogen peroxide for 20 min to eliminate endogenous peroxidase. To retrieve antigen, the sections were incubated in 10 mM sodium citrate buffer (pH 6.0) at 100 °C for 20 min. Then the specimens were washed with 1× PBS three times and incubated with 4-HNE antibody (Abcam, ab46545, 1:200 dilution) at 4 °C overnight. The next day, the specimens were rinsed with 1× PBS three times, followed by applying the ImmPRESS HRP Horse Anti-Rabbit IgG Polymer Detection Kit (MP-7401, Vector laboratory). Finally, ImmPACT DAB Substrate Kit (SK-4105, Vector laboratory) was used to detect the signal. The IHC images were photographed in a microscope (Nikon ECLIPSE Ni) and mean optical density (MOD) values for each specimen were calculated by Image-pro Plus software.

## Xenograft experiments

Randomization of animals was performed before the xenograft experiments. The tumors in the xenograft experiments did not exceed the limit for tumor burden (10% of total body weight or 2 cm in diameter). Cells resuspended with sterile 1× PBS were mixed with Matrigel (Corning, cat#354248) at a 1:1 ratio (volume) and injected subcutaneously into six-week-old female nude mice. For xenograft of A549 and H460 cells, $3.0 \times 10^6$ of sgNC and sgGAS41 cells were used, mice were euthanized with $CO_2$ at 4–5 weeks after injection, and tumors were dissected and weighed. A549 sgNC group and one group of sgGAS41 mice were intraperitoneally injected with saline. Ferrostatin-1 was intraperitoneally injected into the other group of sgGAS41 mice at a dose of 1 mg/kg every day for 14 days, mice were euthanized with $CO_2$, and tumors were dissected and weighed. For xenograft of A549 shGAS41 Tet-on inducible cells, $3.0 \times 10^6$ of A549 shGAS41 Tet-on inducible cells were used. Mice were fed with either regular food or food containing 625 mg/kg doxycycline hyclate (Doxycycline diets, TD08541, Envigo). Four to five weeks after injection, mice were euthanized with $CO_2$, and tumors were dissected and weighed for further experiments.

## Statistics and reproducibility

The *YEAST4* genetic alteration data and the corresponding mRNA alteration associated with *YEAST4* genetic amplification were derived from LSCC (TCGA, PanCancer Atlas) of the cBioportal for Cancer Genomics databases (http://www.cbioportal.org/). The "Survival Plots" module of GEPIA2 web server (http://gepia2.cancer-pku.cn/#analysis) was used to obtain the Kaplan−Meier plots of the combined (*YEAST4*, *SLC7A11*, *GCLC*, and *NQO1*) signature in LUAD, under the settings of group cutoff = Median. The HTSeq-FPKM RNA-seq expression data and clinical data of LUAD were retrieved from The TCGA (https://portal.gdc.cancer.gov). Kaplan−Meier survival curves for overall survival (months) were analyzed in R software (version 3.6.3, http://r-project.org/) using Package 'survminer' and Package 'survival' with Log-rank Mantel−Cox test method.

Two-tailed unpaired Student's *t* test by GraphPad Prism 8.0.2 or Microsoft Excel was done for the statistical analyzes without specific statements to determine *p* values. Data represented in the figures were shown with the error of the mean (mean ± SD) without specific statements. For all tests, $p < 0.05$ was considered statistically significant between groups. Flow cytometry data were analyzed by FlowJo v10. Data were graphed using GraphPad Prism 8.0.2. All experiments were independently repeated three times as stated in Figure Legends. The experimental sample size is indicated in the text and Figure Legends. No data was excluded from the analyzes; data distribution was assumed to be normal, but this was not formally tested. Mice for the xenograft experiments were allocated randomly to each experimental group. For in vitro experiments, experiments design and analysis were conducted by two individual investigators, so two investigators were blinded on group allocation and data analysis, respectively.

## Reporting summary

Further information on research design is available in the Nature Portfolio Reporting Summary linked to this article.

## Data availability

The CUT&RUN data generated in this study have been deposited in the GEO database under accession code GSE256462. All other data needed to evaluate the conclusions in this study are available in the main text and its Supplementary Information. The following public databases were used in this study (see "Methods" for more details): cBioPortal for Cancer Genomics (https://www.cbioportal.org/) and The Cancer Genome Atlas Program (TCGA) (https://www.cancer.gov/ccg/research/genome-sequencing/tcga). All other data and materials are available from the corresponding author upon request. Source data are provided with this paper.

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

## Acknowledgements

This work was supported by the National Cancer Institute of the National Institutes of Health under Award R35CA253059, RO1CA258390, and R01CA254970 to W.G., and award R35CA209896 to B.R.S., and R01CA204232, R01CA258622, and R01CA166413 to X.J. This work was also supported by NCI cancer center core grant P30 CA008748 to Memorial Sloan-Kettering Cancer Center. We acknowledge the support from the Herbert Irving Comprehensive Cancer Center (HICCC; P30 CA13696) and thank the Molecular Pathology and Proteomics of Shared Resources of HICCC. The content is solely the responsibility of the authors and does not necessarily represent the official views of the National Institutes of Health.

## Author contributions

Conception and experimental design: Z.W., X.Y., D.C., and W.G.; CRISPR Screen and analyze data: Z.W., X.Y., Z.L., S.D., and Z.Z.; Cell experiments: Z.W., X.Y., and Y.L.; Animal experiments: Z.W. and X.Y.; Writing, review, and editing manuscript: Z.W., X.Y., X.J., B.R.S., and W.G.; Project supervision: W.G.

## Competing interests

B.R.S. is an inventor on patents and patent applications involving ferroptosis; co-founded and serves as a consultant to ProJenX, Inc. and Exarta Therapeutics; holds equity in Sonata Therapeutics; serves as a consultant to Weatherwax Biotechnologies Corporation and Akin Gump Strauss Hauer & Feld LLP. X.J. is an inventor on patents related to autophagy and cell death; and holds equity of and consults for Exarta Therapeutics and Lime Therapeutics. The other authors declare no competing interests.
