## [Peer Review File · Nature Communications]

GAS41 modulates ferroptosis by anchoring NRF2 on chromatinREVIEWER COMMENTS

Reviewer #1 (Remarks to the Author):

In the presented study, the authors elucidate a novel role for the oncoprotein GAS41 (YEATS4) in tumor development, specifically its function as a significant repressor of ferroptosis. Through comprehensive CRISPR-Cas9 library screenings, the authors identified GAS41 and found that a reduction of GAS41 levels enhances the susceptibility of cancer cells to ferroptosis. Intriguingly, GAS41 was found to be critical for the activation of NRF2 target genes, particularly SLC7A11, which is essential for repressing ferroptosis. The mechanism posited is that GAS41, by recognizing the H3K27-ac marker, facilitates the anchoring of NRF2 to chromatin, thereby promoting its transcriptional activity. This GAS41-NRF2 interaction is shown to be pivotal for the effective binding of NRF2 to target promoters, such as that of SLC7A11. Significantly, *in vivo* experiments using mouse tumor models reinforce the oncogenic relevance of the GAS41-mediated regulation of ferroptosis. In sum, the findings not only underscore GAS41 as a potential therapeutic target for activating ferroptosis to retard tumor growth but also spotlight an unprecedented mechanism of transcriptional regulation via the anchoring of NRF2 on chromatin by GAS41. Overall, the manuscript exhibits considerable depth and significance. However, the specific concerns listed below need to be addressed.

“loss of GAS41 significantly diminishes the ability of NRF2 to bind the SLC7A11 promoter *in vivo*, indicating that the GAS41-NRF2 interaction is critical for the stable binding of NRF2 to chromatin”
NRF2 is known to bind to the ARE in its target genes. Consequently, the outlined sentence is confusing. Clarification is needed on whether the GAS41-NRF2 interaction is essential for the stable binding of NRF2 to the AREs or merely to the chromatin in general.

The term “surprisingly” in the context of “Surprisingly, NRF2 is not required for the recruitment of GAS41 to the promoter of SLC7A11” may seem out of place given GAS41's known role as a histone reader. Considering GAS41's ability to bind to promoters through histone binding, it might be expected that it can be recruited to the promoter of SLC7A11 without the direct involvement of NRF2

Fig. 1D: If cells with sgGAS41 already exhibit reduced SLC7A11 levels, it raises the question as to why these cells still respond to IKE treatment as robustly as they do to TBH, especially when considering that SLC7A11 is the target protein for IKE (comparing panel c with panel d).

Fig. 2. “GAS41 suppresses ferroptosis in a p53-independent manner” should be in the supplemental result part. It is related with the lab's main focus, i.e. p53, however, it is not relevant to this study since the effect is independent of p53.

The initial choice by the author to specifically focus on SLC7A11 in examining the anti-ferroptotic effects of GAS41 might not be immediately clear, particularly when taking into account GAS41's role as a histone acetylation reader. What prompted this specific selection needs further elucidation, since many anti-oxidant proteins, such as GPX4, FSP, and DHODH, have been discovered to control the ferroptosis

process.

“neither loss of AA. 184-227 nor AA. 207-227 within the C-terminal of GAS41 abrogated the binding between NRF2 to GAS41,”

This sentence must be incorrect since this sentence specifies that these regions (AA. 184-227 and AA. 207-227) in the GAS41 protein are not necessary for its interaction with NRF2.

Fig. 4H legend “Luciferase reporter assay of plasmid containing SLC7A11-ARE sequence in A549 sgNC and sgGAS41 cells with same amount of NRF2 expressing vector in dose-dependent manner.”

The term "dose-dependent manner" in the provided figure legend from the authors is ambiguous. It is unclear from the legend alone whether "dose-dependent" refers to varying concentrations of the NRF2 expressing vector, a specific stimulant, or another component entirely. To fully understand the context and implications, readers would benefit from additional information or clarification within the main content of the paper.

The data in Fig. 6d and 6e seem to be at odds, particularly when considering that NRF2 and CBP are likely part of the same protein complex. One would expect the CHIP enrichment for the ARE of GCLC or SLC7A11 to be comparable using anti-Nrf2 or anti-CBP antibodies. Moreover, the experimental conditions between the two figures differ: in 6d, GAS41 is knocked down, while in 6e, both NRF2 and GAS41 are overexpressed. To ensure consistency and clarity in the results, it would be prudent to conduct a CBP CHIP enrichment of the ARE under the exact conditions presented in 6e.

The contents of Fig. 6 and Fig. 7 appear to overlap. These two figures could be merge into one figures with non-essential figures put into the supplemental figure section/

There are typos in Line 427 “indicating that indicate that SLC7A11- and GCLC-mediated GSH”

There are typos in Line 496 “Liu human CRISPR-Cas9 knockout library (H1 and H2) was received”

A comprehensive review for typographical and other errors throughout the manuscript is essential. Figure legends should contain more detailed information to ensure readers can fully understand the presented data.

Reviewer #2 (Remarks to the Author):

In this study, Wang et al. report a role of the histone acetylation reader GAS41 in orchestrating the recruitment of the NRF2 transcription factor onto chromatin in response to oxidative stress. GAS41 was identified through a CRISPR screen as a regulator of ferroptosis in the presence of ROS stress. Furthermore, their findings demonstrate GAS41's interaction with NRF2 and its necessity for NRF2-dependent gene expression. Mechanistically, GAS41 recognizes histone H3K27 acetylation, thereby

facilitating the recruitment and stabilization of NRF2 binding to the promoters of target genes.

While this study offers intriguing insights, several critical points merit consideration.

1.The study primarily focuses on only three target genes (GCLC, NQO1, and SLC7A11). It remains unclear whether this regulation is a widespread mechanism governing NRF2 and GAS41 function in stress response. A comprehensive examination is warranted through genome-wide assessments, including NRF2 and GAS41 ChIP-seq, as well as RNA-seq under conditions identical to those employed in the study (e.g., NRF2 and GAS41 knockout, TBH treatment, etc.).

2.Related to the previous point, are any of these three genes functionally important? Can restoring the expression of any of these genes in GAS41 knockout cells rescue cellular responses to TBH?

3.In the rescue experiments (as presented in Figs. 5 and 6), two truncation mutations of GAS41 are employed to assess the effects of histone acetylation binding and NRF2 binding on NRF2 chromatin recruitment and target gene regulation. However, the use of such extensive truncations may introduce unknown effects on the protein. For instance, it is uncertain whether the truncated proteins are localized within the nucleus or not. To address this, it is recommended that the authors employ point mutations that specifically abolish either histone binding or NRF2 binding in relevant experiments.

4.There appears to be a discrepancy in the abstract regarding the necessity of NRF2 for the recruitment of GAS41 to the promoter of SLC7A11. Fig. 7 clearly indicates a substantial reduction in GAS41 occupancy upon NRF2 knockdown. This contradiction should be addressed and clarified in the abstract.

5.On page 6, the authors suggest that "the effects of GAS41 knockdown in H1299 cells on SLC7A11 and NQO1 mRNA levels were more pronounced under tert-butylhydroquinone (tBHQ) treatment (Fig. 4f,g)." However, the presented data does not seem to support this conclusion.

#RE-NCOMMS-23-42507 (1-10-2024)

Authors' response to Reviewers (first Round)

Authors' response to Reviewer #1

In the presented study, the authors elucidate a novel role for the oncoprotein GAS41 (YEATS4) in tumor development, specifically its function as a significant repressor of ferroptosis. Through comprehensive CRISPR-Cas9 library screenings, the authors identified GAS41 and found that a reduction of GAS41 levels enhances the susceptibility of cancer cells to ferroptosis. Intriguingly, GAS41 was found to be critical for the activation of NRF2 target genes, particularly SLC7A11, which is essential for repressing ferroptosis. The mechanism posited is that GAS41, by recognizing the H3K27-ac marker, facilitates the anchoring of NRF2 to chromatin, thereby promoting its transcriptional activity. This GAS41-NRF2 interaction is shown to be pivotal for the effective binding of NRF2 to target promoters, such as that of SLC7A11. Significantly, in vivo experiments using mouse tumor models reinforce the oncogenic relevance of the GAS41-mediated regulation of ferroptosis. In sum, the findings not only underscore GAS41 as a potential therapeutic target for activating ferroptosis to retard tumor growth but also spotlight an unprecedented mechanism of transcriptional regulation via the anchoring of NRF2 on chromatin by GAS41. Overall, the manuscript exhibits considerable depth and significance. However, the specific concerns listed below need to be addressed.

Overall Response: We greatly appreciate the positive comments on our study by Reviewer #1. We also thank the reviewer for raising several very important points to strengthen this study. As described below, each one of the reviewer's comments has been addressed carefully and we have also modified the manuscript according to the reviewer's suggestions.

Recommended comments from Reviewer#1 and point-to-point responses.

1. The reviewer stated *"loss of GAS41 significantly diminishes the ability of NRF2 to bind the SLC7A11 promoter in vivo, indicating that the GAS41-NRF2 interaction is critical for the stable binding of NRF2 to chromatin"* NRF2 is known to bind to the ARE in its target genes. Consequently, the outlined sentence is confusing. Clarification is needed on whether the GAS41-NRF2 interaction is essential for the stable binding of NRF2 to the AREs or merely to the chromatin in general.

Response: The reviewer was right; GAS41-mediated regulation of NRF2 binding on chromatin acts through the histone modification by recognizing the H3K27-ac marker. Thus, GAS41 is not required for NRF2 to recognize the ARE sites on the promoters; instead, plays a critical role for its stable binding with the chromatin. We appreciate the comments of the reviewer and we have now modified the abstract and the text accordingly.

2. The reviewer stated *The term "surprisingly" in the context of "Surprisingly, NRF2 is not required for the recruitment of GAS41 to the promoter of SLC7A11" may seem out of place given GAS41's known role as a histone reader. Considering GAS41's ability to bind to promoters through histone binding,*

it might be expected that it can be recruited to the promoter of SLC7A11 without the direct involvement of NRF2.

Response: The reviewer is right; through acting as a histone acetylation reader, GAS41 is able to bind the promoters of the chromatin through histone binding. We have removed “surprisingly” in the revised manuscript. We sincerely thank Reviewer#1 for the suggestion to avoid any confusion.

3. The reviewer stated *“Fig. 1D: If cells with sgGAS41 already exhibit reduced SLC7A11 levels, it raises the question as to why these cells still respond to IKE treatment as robustly as they do to TBH, especially when considering that SLC7A11 is the target protein for IKE (comparing panel c with panel d).”*

Response: This is an interesting point. First, the reviewer is right; IKE, an erastin analog, should mainly target SLC7A11 (PMID: 30799221)¹(PMID: 22632970)². Nevertheless, even in the cells with low levels of SLC7A11, this compound remains effective to promote ferroptosis. For example, Dixon et al. found that knockdown of SLC7A11 sensitized cells to erastin-induced ferroptosis (PMID: 22632970)². Moreover, Zhang et al. showed that downregulation of SLC7A11 modulated by BAP1 sensitized cells to erastin-induced ferroptosis (PMID: 30202049)³. Second, IKE/erastin-induced ferroptosis can also act through a SLC7A11-independent manner. For example, Wu et al. found that erastin can cause CMA-mediated degradation of GPX4 to induce ferroptosis (PMID: 30718432)⁴. A significant decrease in protein levels of GPX4 was also reported upon erastin treatment (PMID: 33707434)⁵. Thus, the ferroptosis induced by erastin/IKE may also act through modulating GPX4 levels. Finally, our study found that GAS41 is a critical modulator for GSH metabolism through regulating GCLC levels, in addition to its effect on SLC7A11. Thus, it is not surprising that IKE sensitizes GAS41 knockout/knockdown cells to ferroptosis. We appreciate the excellent points raised by the reviewer.

4. The reviewer stated *“Fig. 2. “GAS41 suppresses ferroptosis in a p53-independent manner” should be in the supplemental result part. It is related with the lab’s main focus, i.e. p53, however, it is not relevant to this study since the effect is independent of p53.”*

Response: The reviewer’s suggestion is well taken. We have moved the original Fig.2 “GAS41 suppresses ferroptosis in a p53-independent manner” to Supplementary Fig.2 in the revised manuscript.

5. The reviewer stated *“the initial choice by the author to specifically focus on SLC7A11 in examining the anti-ferroptotic effects of GAS41 might not be immediately clear, particularly when taking into account GAS41’s role as a histone acetylation reader. What prompted this specific selection needs further elucidation, since many anti-oxidant proteins, such as GPX4, FSP, and DHODH, have been discovered to control the ferroptosis process.”*

Response: This is an excellent point; we apologize that we did not explain our rationality for investigating the SLC7A11 regulation very clearly. Indeed, we examined the possibility of GAS41 in regulating a number of well-known factors in

ferroptosis defense, including GPX4, FSP1, and DHODH. As is shown in **Fig. R1**, unlike SLC7A11, the loss of GAS41 had no obvious effect on the expression of GPX4, FSP1, and DHODH. Thus, SLC7A11 is likely the major target to be significantly regulated upon loss of GAS41 in ferroptosis defense. We appreciate the comments of the reviewer and we have now modified the text accordingly.

Fig. R1 for Reviewer#1

Western blot analysis of the protein levels of FSP1, GPX4, and DHODH in sgNC, sgGAS41#1, and sgGAS41#2 A549 cells.

6. The reviewer stated *“neither loss of AA. 184-227 nor AA. 207-227 within the C-terminal of GAS41 abrogated the binding between NRF2 to GAS41,”* This sentence must be incorrect since this sentence specifies that these regions (AA. 184-227 and AA. 207-227) in the GAS41 protein are not necessary for its interaction with NRF2.

Response: The reviewer was right; we apologize for the error. We have now corrected this sentence as follows: *“Loss of AA. 184-227 or 207-227 within the C-terminal of GAS41 both abrogated the binding between NRF2 to GAS41”*. Thanks again for pointing out this point.

7. The reviewer stated *Fig. 4H legend “Luciferase reporter assay of plasmid containing SLC7A11-ARE sequence in A549 sgNC and sgGAS41 cells with same amount of NRF2 expressing vector in dose-dependent manner.”* The term “dose-dependent manner” in the provided figure legend from the authors is ambiguous. It is unclear from the legend alone whether “dose-dependent” refers to varying concentrations of the NRF2 expressing vector, a specific stimulant, or another component entirely. To fully understand the context and implications, readers would benefit from additional information or clarification within the main content of the paper.

Response: The reviewer was right; we apologize for the ambiguous description in figure legends. To avoid any confusion, we have now modified the legend of **Fig. 3h**, as follows: *“Luciferase activity measured in A549 sgNC and sgGAS41 cells transfected with reporter plasmid containing SLC7A11 promoter region bound by NRF2 alone or co-transfected with reporter plasmid containing SLC7A11 promoter region bound by NRF2 and*

dose-dependent overexpression of NRF2 expressing vector. Renilla control reporter was used as a transfection internal control.”

8. The reviewer stated *The data in Fig. 6d and 6e seem to be at odds, particularly when considering that NRF2 and CBP are likely part of the same protein complex. One would expect the ChIP enrichment for the ARE of GCLC or SLC7A11 to be comparable using anti-Nrf2 or anti-CBP antibodies. Moreover, the experimental conditions between the two figures differ: in 6d, GAS41 is knocked down, while in 6e, both NRF2 and GAS41 are overexpressed. To ensure consistency and clarity in the results, it would be prudent to conduct a CBP ChIP enrichment of the ARE under the exact conditions presented in 6e.*

Response: This is an excellent suggestion. The reviewer was right; in the last version, we only showed the CBP recruitment in GAS41-null cells, indicating that CBP enrichment is not affected by GAS41. To further validate this important point, we have now also performed CBP enrichment under the conditions that NRF2 and GAS41 are overexpressed according to the reviewer’s suggestion. As shown in **Fig. R2**, the ChIP-qPCR analysis revealed that GAS41 expression has no obvious effect on the recruitment of CBP to the promoter of *SLC7A11* ($p=0.339$). Similar results were also obtained from the promoter of *GCLC* (**Fig. R2**). Taken together, these data further validate that GAS41-mediated function may not act through the recruitment of CBP to the promoters of *SLC7A11* or *GCLC*.

Fig. R2 for Reviewer#1

ChIP-qPCR analysis of CBP binding on *SLC7A11* and *GCLC* promoter region in A549 sgGAS41 cells transfected with NRF2 along with vector or GAS41 WT.

9. The reviewer stated *The contents of Fig. 6 and Fig. 7 appear to overlap. These two figures could be merge into one figures with non-essential figures put into the supplemental figure section/*

Response: Following the suggestion by the reviewer, we have combined **Fig.6** and **Fig.7** into one figure as the **new Fig.6** in the revised manuscript.

10. The reviewer stated *There are typos in Line 427 “indicating that indicate that SLC7A11- and GCLC-mediated GSH”*

Response: We have corrected the typos.

11. The reviewer stated *There are typos in Line 496 “Liu human CRISPR-Cas9 knockout library (H1 and H2) was received”*

Response: We have corrected the typos.

12. The reviewer stated *A comprehensive review for typographical and other errors throughout the manuscript is essential. Figure legends should contain more detailed information to ensure readers can fully understand the presented data.*

Response: We apologize for some confusions caused by typos and errors. We have double-checked the words thoroughly and polished the manuscript according to the reviewer's suggestion. We also added more details for the Figure legends to avoid any misunderstanding.

Reference

1. Zhang, Y. et al. Imidazole ketone erastin induces ferroptosis and slows tumor growth in a mouse lymphoma model. *Cell chemical biology* **26**, 623-633 (2019). PMID: 30799221
2. Dixon, S.J. et al. Ferroptosis: an iron-dependent form of nonapoptotic cell death. *Cell*, **149**, 1060-1072 (2012). PMID: 22632970
3. Zhang, Y. et al. BAP1 links metabolic regulation of ferroptosis to tumour suppression. *Nature cell biology*, **20**, 1181-1192 (2018). PMID: 30202049
4. Wu, Z. et al. Chaperone-mediated autophagy is involved in the execution of ferroptosis. *Proceedings of the National Academy of Sciences*, **116**, 2996-3005 (2019). PMID: 30718432
5. Zhang, Y. et al. mTORC1 couples cyst(e)ine availability with GPX4 protein synthesis and ferroptosis regulation. *Nature communications*, **12**, 1589. (2021). PMID: 33707434

Authors' response to Reviewer #2

In this study, Wang et al. report a role of the histone acetylation reader GAS41 in orchestrating the recruitment of the NRF2 transcription factor onto chromatin in response to oxidative stress. GAS41 was identified through a CRISPR screen as a regulator of ferroptosis in the presence of ROS stress.

Furthermore, their findings demonstrate GAS41's interaction with NRF2 and its necessity for NRF2-dependent gene expression. Mechanistically, GAS41 recognizes histone H3K27 acetylation, thereby facilitating the recruitment and stabilization of NRF2 binding to the promoters of target genes.

While this study offers intriguing insights, several critical points merit consideration.

Overall Response: We appreciate the reviewer's thoughtful comments. We have taken all the issues raised from the Reviewer #2 very seriously and addressed each point with detailed explanation and a large amount of new data as shown below (the new data are also incorporated in the revised manuscript). The manuscript is significantly modified according to the reviewer's suggestions.

Recommended comments from Reviewer #2 and point-to-point responses.

1. The reviewer stated *The study primarily focuses on only three target genes (GCLC, NQO1, and SLC7A11). It remains unclear whether this regulation is a widespread mechanism governing NRF2 and GAS41 function in stress response. A comprehensive examination is warranted through genome-wide assessments, including NRF2 and GAS41 ChIP-seq, as well as RNA-seq under conditions identical to those employed in the study (e.g., NRF2 and GAS41 knockout, TBH treatment, etc.).*

Response: We appreciate the constructive comments from the reviewer. We agree that a comprehensive genome-wide analysis will further validate this novel regulatory mechanism between NRF2 and GAS41.

Following the suggestion from the reviewer, we have taken this issue seriously and performed new experiments to address this issue. First, since the ChIP-quality antibody of NRF2 is commercially available, we performed the ChIP-seq analysis of NRF2 in those cancer cells; however, the commercially available GAS41 antibodies are not good enough for ChIP-seq analysis of endogenous GAS41 protein (based on the literature as well as our own experience) (e.g. **PMID: 37311463**)¹. To overcome this technical issue, we decided to perform the ChIP analysis of endogenous NRF2 protein in both native A549 human lung cancer cells and isogenic GAS41-null A549 cells to examine whether the DNA binding activities of endogenous NRF2 on specific target promoters are indeed regulated by loss of GAS41. Moreover, Cleavage Under Targets and Release Using Nuclease (CUT&RUN) technique is well established as a more quantitative assay to monitor the binding activity of transcriptional factors than the traditional ChIP-seq assay (**PMID: 28079019**)². To this end, the CUT&RUN assays were performed by using control A549 cells and GAS41-null A549 cells to analyze the effect of GAS41 on NRF2 DNA binding ability, followed by next-generation sequencing.

As shown in **Fig. R1a**, as expected, high levels of NRF2 binding on chromatin were detected by CUT&RUN analysis; however, loss of GAS41 resulted in a significant global decrease in NRF2 binding on chromatin. Indeed, consistent with

the ChIP-qPCR results that we obtained before, NRF2 binding activities on the promoters of *SLC7A11*, *GCLC*, and *NQO1* were significantly reduced upon loss of GAS41 expression (**Fig. R1b-d**) (**Figures incorporated into Fig. 5e, f and Supplementary Fig. 6f**). Moreover, based on the CUT&RUN analysis, we identified several new promoters of NRF2 where the NRF2 binding activities are also regulated by GAS41. Those new targets include *AKR1B10*, *AKR1C2*, *GCLM*, *SRXN1*, *G6PD*, and *AKR1C3* (**Fig. R1e-j**). In addition, we also found the GAS41 was not involved in some NRF2 targets' regulation, such as *ME1* and *ATG7* (**Fig. R1k, l**) (**Figure incorporated into Supplementary Fig. 6g**), suggesting that GAS41-mediated regulation of NRF2 is promoter-specific.

Finally, to further validate the above results, we performed RNA-seq assays of A549 shNRF2 tetracycline-controlled (tet-on) inducible cells incubated without or with doxycycline (0.2 µg/mL) for 72h, A549 shGAS41 tet-on-inducible cells with doxycycline (0.2 µg/mL) for 72h, and A549 cells without or with TBH-treated (120 µM) for 6h. As shown in **Fig. R1m**, as expected, the levels of *SLC7A11*, *GCLC*, and *NQO1*, were dramatically reduced upon GAS41 knockdown. Consistent with the above CUT&RUN analysis, all six new targets identified above are also significantly regulated by the status of GAS41. Notably, as expected, the levels of *ME1* and *ATG7* mRNA were significantly reduced upon NRF2 knockdown as both are transcription targets of NRF2; however, in contrast to the levels of *SLC7A11* or other newly identified targets, GAS41 knockdown did not dramatically reduce the levels on *ME1* or *ATG7* (**Fig. R1m**), indicating that GAS41 regulates NRF2 transcriptional activation in a promoter-specific manner. Moreover, we also analyzed the effects of TBH on those targets; as shown in **Fig. R1n**, although some very modest effects on were observed for some targets, overall, the TBH treatment does not dramatically affect the expression levels of those NRF2 targets, consistent with the notion that the major function of the TBH treatment is to increase the levels of ROS.

Taken together, although further analysis of these new the CUT&RUN assays and RNA-seq-assays were clearly required, these genome-wide analyses indicate that GAS41 is able to regulate NRF2-mediated transcriptional activation of its cellular targets in a promoter-specific manner thus further validating our working model.

Fig. R1 for Reviewer#2

a Heatmap of NRF2 CUT&RUN signal at each of 3105 peaks found in A549 sgNC and sgGAS41.

b-l Snapshot of NRF2 CUT&RUN signal at 11 target genes loci, including *SLC7A11* (**b**), *GCLC* (**c**), *NQO1* (**d**), *AKR1B10* (**e**), *AKR1C2* (**f**), *GCLM* (**g**), *SRXN1* (**h**), *G6PD* (**i**), *AKR1C3* (**j**), *ME1* (**k**), and *ATG7* (**l**).

m Heatmap of 11 downregulated known NRF2-target genes from RNA-seq of A549 shNRF2 tet-on inducible cells incubated without or with doxycycline (0.2 µg/mL) for 72h and A549 shGAS41 tet-on inducible cells with doxycycline (0.2 µg/mL) for 72h, autoscaled to z scores, and coded blue (low values) to red (high values).

n Heatmap of 9 known NRF2-target genes downregulated by GAS41 knockdown from RNA-seq of A549 cells treated with DMSO or TBH (120 μ M) for 6h, autoscaled to z scores, and coded blue (low values) to red (high values).

2. The reviewer stated *Related to the previous point, are any of these three genes functional important? Can restoring the expression of any of these genes in GAS41 knockout cells rescue cellular responses to TBH?*

Response: This is an excellent point. These three targets are well known to play a critical role in regulating ferroptosis. For example, GCLC and SLC7A11 are crucial for glutathione (GSH) synthesis: Cystine is transported into cells by system x_c^- (SLC7A11), and GCLC catalyzes the first step of GSH synthesis by acting as ligase of cysteine and glutamate. GSH, the natural antioxidant, is a cofactor for multiple antioxidant proteins, including GSH peroxidase (GPX4), to maintain redox homeostasis. SLC7A11 and GCLC have a well-established role in ferroptosis protection and tumor development in GSH-dependent and GSH-independent manner (**PMID: 25799988**; **PMID: 33000412**; **PMID: 33357455**)³⁻⁵. Last, NQO1 catalyzes the two-electron mediated reduction of quinone (Coenzyme Q10) to hydroquinone, which can protect cellular membranes from ferroptotic stress. NQO1 knockdown increased growth suppression upon treating ferroptosis inducers in hepatocellular carcinoma (**PMID: 26403645**)⁶. Altogether, these three genes are all functionally important for antioxidant defense.

As suggested by the reviewer, since the levels of these three targets are downregulated by loss of GAS41, we examine whether ectopically expression of SLC7A11, GCLC, or NQO1 in GAS41-null cells is able to modulate ferroptotic response. As shown in **Fig.R2a, b**, the levels of SLC7A11 were significantly upregulated upon ectopic expression of SLC7A11 in GAS41-null cells; indeed, the levels of ferroptosis were also partially reduced. Similar experiments were also performed for GCLC and NQO1 in GAS41-null cells (**Fig.R2c-f**). Again, the levels of ferroptosis were also partially reduced upon ectopic expression of GCLC or NQO1. These results not only validate the effects of GCLC, NQO1, and SLC7A11 in modulating ferroptosis but also suggest that GAS41-mediated effect in ferroptosis acts, at least in part, through these three important targets.

Fig. R2 for Reviewer#2

a Western blot analysis of SLC7A11 protein levels of A549 sgNC re-expressed with vector, sgGAS41 cells re-expressed with vector or SLC7A11 expression plasmid.

b Cell death of A549 sgNC re-expressed with vector, sgGAS41 cells re-expressed with vector or SLC7A11 expression plasmid under TBH treatment (120 μ M) for 6h.

c Western blot analysis of GCLC protein levels of A549 sgNC re-expressed with vector, sgGAS41 cells re-expressed with vector or GCLC expression plasmid.

d Cell death of A549 sgNC re-expressed with vector, sgGAS41 cells re-expressed with vector or GCLC expression plasmid under TBH treatment (120 μ M) for 6h.

e Western blot analysis of NQO1 protein levels of A549 sgNC re-expressed with vector, sgGAS41 cells re-expressed with vector or NQO1 expression plasmid.

f Cell death of A549 sgNC re-expressed with vector, sgGAS41 cells re-expressed with vector or NQO1 expression plasmid under TBH treatment (120 μ M) for 6h.

3. The reviewer stated “*In the rescue experiments (as presented in Figs. 5 and 6), two truncation mutations of GAS41 are employed to assess the effects of histone acetylation binding and NRF2 binding on*

NRF2 chromatin recruitment and target gene regulation. However, the use of such extensive truncations may introduce unknown effects on the protein. For instance, it is uncertain whether the truncated proteins are localized within the nucleus or not. To address this, it is recommended that the authors employ point mutations that specifically abolish either histone binding or NRF2 binding in relevant experiments.”

Response: This is an excellent point raised by the reviewer. Following the reviewer’s suggestion, we employed GAS41 point mutations that specifically abrogated the binding with histone acetylation or NRF2, respectively, to avoid the unknown effects on the GAS41 protein introduced by truncations. Based on existing results, AA. 1-160 or AA. 207-227 was necessary for histone acetylation-GAS41 binding or NRF2-GAS41 interaction, respectively (**Fig. R3a**). Thus, we examined the interaction between NRF2 and several GAS41 point mutants located in AA. 207-227, including L211A, K212Q, E214A, R216T, K217Q, and L218A. Indeed, one point mutant of GAS41 (L211A) dramatically abolished the binding with NRF2, compared with GAS41 wildtype (WT) (**Fig.R3b**). In addition, Hsu et al. have reported that GAS41 W93A mutants lost the ability to bind with histone acetylation (**PMID: 29437725**)⁷. We further found these mutants retain their ability to interact with NRF2 (**Fig.R3b**). Thus, based on these new screening assays, we have now obtained two important functional point mutants: the point mutant of GAS41 (L211A) is specifically defective for NRF2 binding but not H3-K27-ac whereas the point mutant of GAS41 (W93A) is defective for H3-K27-ac but not NRF2.

Next, GAS41 WT, GAS41-W93A, and GAS41-L211A were expressed in GAS41-null A549 cells to dissect the effect of GAS41 on modulating NRF2 transcriptional functions. As shown in **Fig. R3c**, western blot analysis revealed that GAS41 WT, but not either H3-K27-ac binding deficient mutant (W93A) or NRF2 binding-deficient mutant (L211A), is able to restore the ability of NRF2-mediated transcriptional activation of *NQO1*, *SLC7A11*, and *GCLC* in GAS41-null cells. Similar results were also obtained for the mRNA levels of those NRF2-targets by qPCR-analysis (**Fig. R3d**). Moreover, GAS41 WT but not GAS41-W93A, or GAS41-L211A, is able to increase the GSH levels in GAS41-null cells (**Fig. R3e**), indicating that both the GAS41-NRF2 binding, and the GAS41-H3-K27-ac interaction are required for NRF2 transcriptional activation mediated by GAS41. Consistent with these observations, the CHIP-qPCR analysis showed that GAS41 WT but not GAS41-W93A, or GAS41-L211A, is able to increase the binding of NRF2 to the promoters of *SLC7A11* and *GCLC* in GAS41-null cells (**Fig. R3f**).

Finally, we have also performed ferroptosis assays with those points’ mutant. As expected, GAS41 WT but not GAS41-W93A, or GAS41-L211A, is able to suppress ferroptosis in GAS41-null cells (**Fig. R3g**). In parallel, lipid peroxidation induced by GAS41 deficiency was downregulated by overexpressed GAS41 WT, but not GAS41 W93A or L211A mutants (**Fig. R3h, i**). Taken together these new data further validate that both the GAS41-NRF2 binding, and the GAS41-H3-K27-ac interaction are critical for GAS41-mediated effects on NRF2 transactivation and ferroptosis defense. We appreciated the excellent suggestion by the reviewer#2. Accordingly, this part of the data has now been incorporated into the revised manuscript as new versions of **Fig.4** and **Fig.5c**.

Fig. R3 for Reviewer#2

a Schematic diagram of the GAS41 domains and GAS41 mutants. GAS41- amino acids 1-206 contains YEATS domain, A box and partial coiled-coil motif, referred to AA. 1-206; GAS41- amino acids 161-227 contains complete coiled-coil motif, referred to AA. 161-227; GAS41-W93A contains a point mutation at position 93 tryptophan (W) which was mutated to alanine (A); GAS41-L211A contains a point mutation at position 211 leucine (L) which was mutated to alanine (A).

b Western blot analysis of interaction between NRF2 and GAS41 mutants (W93A and L211A) in HEK293T cells.

c Western blot analysis of SLC7A11, GCLC, and NQO1 protein levels of A549 sgNC re-expressed with vector, sgGAS41 cells re-expressed with vector, GAS41 WT, GAS41-W93A mutant, or GAS41-L211A mutants.

d RT-qPCR analysis of *SLC7A11*, *GCLC*, and *NQO1* mRNA levels of A549 sgNC re-expressed with vector, sgGAS41 cells re-expressed with vector, GAS41 WT, GAS41-W93A mutant, or GAS41-L211A mutants.

e Measurement of GSH concentration of A549 sgNC re-expressed with vector, sgGAS41 cells re-expressed with vector, GAS41 WT, GAS41-W93A mutant, or GAS41-L211A mutants.

f CHIP-qPCR analysis of overexpressed NRF2 binding on *SLC7A11* and *GCLC* promoter region in A549 sgGAS41 cells transfected with NRF2 expressing vector along with vector, GAS41 WT, GAS41 W93A, or GAS41 L211A.

g Cell viability of A549 sgNC re-expressed with vector, sgGAS41 cells re-expressed with vector, GAS41 WT, GAS41-W93A mutant, or GAS41-L211A mutants treated with TBH (120 μ M, **left panel**) for 6h or IKE (3 μ M, **right panel**) for 24h.

h, i Assessment of lipid peroxidation (**h**) and statistical bar graph (**i**) by flow cytometry after C11-BODIPY staining of A549 sgNC re-expressed with vector, sgGAS41 cells re-expressed with vector, GAS41 WT, GAS41-W93A mutant, or GAS41-L211A mutants upon TBH treatment (100 μ M) for 4h.

4. The reviewer stated *“there appears to be a discrepancy in the abstract regarding the necessity of NRF2 for the recruitment of GAS41 to the promoter of SLC7A11. Fig. 7 clearly indicates a substantial reduction in GAS41 occupancy upon NRF2 knockdown. This contradiction should be addressed and clarified in the abstract.*

Response: The reviewer was right; in addition to GAS41-mediated regulation of NRF2 binding on chromatin, NRF2 also plays a significant role in modulating the levels of GAS41 binding on chromatin. Thus, GAS41 and NRF2 are reciprocally regulated by their interactions, resulting in the enhancement of both proteins to interact with chromatin. We appreciate the comments of the reviewer and we have now modified the abstract and the text accordingly.

5. The reviewer stated *On page 6, the authors suggest that “the effects of GAS41 knockdown in H1299 cells on SLC7A11 and NQO1 mRNA levels were more pronounced under tert-butylhydroquinone (tBHQ) treatment (Fig. 4f,g).” However, the presented data does not seem to support this conclusion.*

Response: The reviewer was right; the effects of GAS41 knockdown in H1299 cells on *SLC7A11* and *NQO1* mRNA levels were observed under both untreated conditions and the conditions after the tert-butylhydroquinone (tBHQ) treatment. We appreciate the comments of the reviewer and we have now modified the text accordingly.

Reference

1. Liu, N. Histone H3 lysine 27 crotonylation mediates gene transcriptional repression in chromatin. *Molecular Cell*, **83**, 2206-2221. (2023). PMID: 37311463.
2. Skene, P.J. and Henikoff, S. An efficient targeted nuclease strategy for high-resolution mapping of DNA binding sites. *elife* **6**, e21856 (2017). PMID: 28079019
3. Jiang, L., Kon, N., Li, T., Wang, S.J., Su, T., Hibshoosh, H., Baer, R. and Gu, W. Ferroptosis as a p53-mediated activity during tumour suppression. *Nature*, **520**, 57-62 (2015). PMID: 25799988
4. Koppula, P., Zhuang, L. and Gan, B. Cystine transporter SLC7A11/xCT in cancer: ferroptosis, nutrient dependency, and cancer therapy. *Protein & cell*, **12**, 599-620 (2021). PMID: 33000412
5. Kang, Y.P., Mockabee-Macias, A., Jiang, C., Falzone, A., Prieto-Farigua, N., Stone, E., Harris, I.S. and DeNicola, G.M. Non-canonical glutamate-cysteine ligase activity protects against ferroptosis. *Cell metabolism*, **33**, 174-189 (2021). PMID: 33357455
6. Sun, X., Ou, Z., Chen, R., Niu, X., Chen, D., Kang, R. and Tang, D. Activation of the p62-Keap1-NRF2 pathway protects against ferroptosis in hepatocellular carcinoma cells. *Hepatology*, **63**, 173-

184 (2016). PMID: 26403645

7. Hsu, C.C., Shi, J., Yuan, C., Zhao, D., Jiang, S., Lyu, J., Wang, X., Li, H., Wen, H., Li, W. and Shi, X. Recognition of histone acetylation by the GAS41 YEATS domain promotes H2A. Z deposition in non-small cell lung cancer. *Genes & development*, **32**, 58-69 (2018). PMID: 29437725

REVIEWERS' COMMENTS

Reviewer #2 (Remarks to the Author):

The authors have answered most of my questions with additional experiments. The new results are important evidence supporting the main conclusions of the paper and therefore should be included in the main figures with NGS data deposited properly.

Reviewer #3 (Remarks to the Author):

In this revised manuscript, the authors report the discovery of GAS41 as an endogenous suppressor of ferroptosis. The manuscript also reveals a novel mechanism of action for GAS41, involving its ability to bridge the interaction between NRF2 and the H3K27 histone acetylation marker, anchoring NRF2 on chromatin and facilitating transcriptional activation of SLC7A11 and other downstream cytoprotective targets of NRF2. Since GAS41 is frequently overexpressed in tumors, interference with GAS41 activity may provide a new avenue for cancer treatment. Thus, the findings in this manuscript are important for both their mechanistic insights and their implications for future clinical application.

In this revision, the authors have responded thoroughly to issues raised during the first round of review. In particular, clarifications and additional experiments requested by Reviewer 1 have been provided regarding the following points: the role of GAS41 in chromatin binding (points 1 and 2); the ability of GAS41 knockdown to sensitize cells to IKE, a ferroptosis-inducing agent that targets SLC7A11, in cells with low endogenous levels of SLC7A11 (point 3); and the rationale for focus on SLC7A11 (point 5). In addition, correction of some typographical errors (point 6), and a ChIP analysis demonstrating lack of CBP enrichment on SLC7A11 and GCLC promoters (point 8), have been provided. Points raised by Reviewer 2 have also been well addressed, notably the identification of specific domains involved in the NRF2-GAS41 interaction using newly-generated point mutations.

Minor typographical errors remain, despite the authors' corrections of errors pointed out by reviewer 1 (points 7 and 10-12). In particular, the phrasing of the legend to 3h ("...bound by NRF2 alone or co-transfected with reporter plasmid containing SLC7A11 promoter region bound by NRF2 and dose-dependent overexpression...") is still confusing. This should be rewritten. Line 302, "neither" should be replaced with "either", etc. The manuscript merits a careful and thorough review for linguistic problems which, though minor, distract attention from the main points of the narrative.

Two points that remain (somewhat surprisingly) unaddressed in this manuscript are the specificity of GAS41 effects – i.e. does it also modulate apoptotic pathways? In addition, the ability of GAS41 knockdown to potentiate the response of tumors to ferroptosis inducers in vivo is not addressed. This seems an unexpected omission given the authors' statement in the discussion that "... the combination

of FAS41 YEATS domain inhibitor with ferroptosis sensitizers could be a promising therapeutic strategy for clinical NSCLC therapy.” This is a prediction that could be easily tested in mice.

#RE-NCOMMS-23-42507A

Authors' response to Reviewers (second round)

Authors' response to Reviewer #2

Reviewer stated: *"The authors have answered most of my questions with additional experiments. The new results are important evidence supporting the main conclusions of the paper and therefore should be included in the main figures with NGS data deposited properly."*

Response: We appreciate the very positive comments on our revised manuscript. Following the suggestion of the reviewer, we have now addressed the minor issue by incorporating the CUT&RUN data into the main figure (the revised Fig. 5) and deposited the NGS raw data onto GEO database (GSE256462).

Authors' response to Reviewer #3

1.Reviewer stated: *“ In this revision, the authors have responded thoroughly to issues raised during the first round of review. In particular, clarifications and additional experiments requested by Reviewer 1 have been provided regarding the following points: the role of GAS41 in chromatin binding (points 1 and 2); the ability of GAS41 knockdown to sensitize cells to IKE, a ferroptosis-inducing agent that targets SLC7A11, in cells with low endogenous levels of SLC7A11 (point 3); and the rationale for focus on SLC7A11 (point 5). In addition, correction of some typographical errors (point 6), and a ChIP analysis demonstrating lack of CBP enrichment on SLC7A11 and GCLC promoters (point 8), have been provided. Points raised by Reviewer 2 have also been well addressed, notably the identification of specific domains involved in the NRF2-GAS41 interaction using newly-generated point mutations.*

Response: We greatly appreciate the positive comments on our study by Reviewer#3.

2. Reviewer stated: *Minor typographical errors remain, despite the authors' corrections of errors pointed out by reviewer 1 (points 7 and 10-12). In particular, the phrasing of the legend to 3h (“...bound by NRF2 alone or co-transfected with reporter plasmid containing SLC7A11 promoter region bound by NRF2 and dose-dependent overexpression....”) is still confusing. This should be rewritten. Line 302, “neither” should be replaced with “either”, etc. The manuscript merits a careful and thorough review for linguistic problems which, though minor, distract attention from the main points of the narrative.*

Response: We have modified the figure legend as the reviewer suggested and corrected the typos (which is highlighted in the revised manuscript).

3. Reviewer stated: *“Two points that remain (somewhat surprisingly) unaddressed in this manuscript are the specificity of GAS41 effects – i.e. does it also modulate apoptotic pathways? In addition, the ability of GAS41 knockdown to potentiate the response of tumors to ferroptosis inducers in vivo is not addressed. This seems an unexpected omission given the authors' statement in the discussion that “... the combination of FAS41 YEATS domain inhibitor with ferroptosis sensitizers could be a promising therapeutic strategy for clinical NSCLC therapy.” This is a prediction that could be easily tested in mice.”*

Response: We appreciated the excellent comments. Although we agree that these are potentially important issues that require new experiments from additional studies of tumor mouse models, these new issues are clearly out of the scope of this study. Nevertheless, following the suggestion by the editor, we have discussed these issues in the revised discussion section (which is highlighted in the revised manuscript).